# BioX-Bridge: Model Bridging for Unsupervised Cross-Modal Knowledge Transfer across Biosignals

**Chenqi Li**\*, **Yu Liu**\*, **Timothy Denison, Tingting Zhu**
Department of Engineering Science
University of Oxford
`{chenqi.li,yu.liu,timothy.denison,tingting.zhu}@eng.ox.ac.uk`

## Abstract

Biosignals offer valuable insights into the physiological states of the human body. Although biosignal modalities differ in functionality, signal fidelity, sensor comfort, and cost, they are often intercorrelated, reflecting the holistic and interconnected nature of human physiology. This opens up the possibility of performing the same tasks using alternative biosignal modalities, thereby improving the accessibility, usability, and adaptability of health monitoring systems. However, the limited availability of large labeled datasets presents challenges for training models tailored to specific tasks and modalities of interest. Unsupervised cross-modal knowledge transfer offers a promising solution by leveraging knowledge from an existing modality to support model training for a new modality. Existing methods are typically based on knowledge distillation, which requires running a teacher model alongside student model training, resulting in high computational and memory overhead. This challenge is further exacerbated by the recent development of foundation models that demonstrate superior performance and generalization across tasks at the cost of large model sizes. To this end, we explore a new framework for unsupervised cross-modal knowledge transfer of biosignals by training a lightweight bridge network to align the intermediate representations and enable information flow between foundation models and across modalities. Specifically, we introduce an efficient strategy for selecting alignment positions where the bridge should be constructed, along with a flexible prototype network as the bridge architecture. Extensive experiments across multiple biosignal modalities, tasks, and datasets show that BioX-Bridge reduces the number of trainable parameters by 88–99% while maintaining or even improving transfer performance compared to state-of-the-art methods. Our code is available at: `https://github.com/chenqi-li/BioX-Bridge`.

## 1 Introduction

Biosignals, such as electrocardiogram (ECG), electroencephalogram (EEG), and photoplethysmography (PPG), provide critical insights into the underlying physiological states of individuals. They are essential tools in modern healthcare and have often been considered the gold standard for diagnostics (Rosenberg & Van Hout, 2013; Stracina et al., 2022). In the past decade, the advancement of artificial intelligence (AI) has enabled remarkable capabilities in automated diagnostics and monitoring, such as stress assessment (Mentis et al., 2024), sleep stage classification (Mostafa et al., 2019), and arrhythmia detection (Parvaneh et al., 2019). However, many biosignal sensors are not suitable for use outside clinical settings due to factors such as user discomfort, high manufacturing costs, and excessive power consumption.

A promising direction is to harness the correlations between different biosignal modalities and perform the same tasks using alternative modalities, making health monitoring systems more accessible, practical, and flexible (Wang et al., 2023; Yang et al., 2023). For example, being able to perform the same tasks using single-lead PPG data from a wearable smartwatch, instead of relying on 12-lead

---

\*Equal contribution.

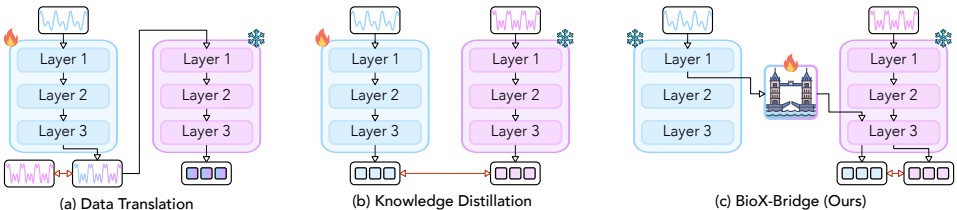

Figure 1: Comparison of unsupervised cross-modal knowledge transfer methods for biosignals. The red arrow indicates loss computation.

ECGs, would greatly reduce hardware complexity and cost, while enabling continuous, user-friendly monitoring in everyday environments. Unfortunately, training such models requires large-scale labeled datasets, which are often difficult to obtain in biosignal applications due to the high cost and domain-specific expertise required for data collection and annotation. This highlights the need for effective knowledge transfer between biosignal modalities, leveraging models trained in old or well-established modalities to support the development of models for new or underrepresented biosignal modalities.

Unsupervised cross-modal knowledge transfer stands out as a practical solution to address the aforementioned needs. Existing methods can be divided into two categories: data translation and knowledge distillation. As illustrated in Figure 1(a), data translation directly translates data from the new modality to the old modality, enabling the direct reuse of existing models from the old modality (Sarkar & Etemad, 2021). However, the exploration of data translation has been limited to a certain pair of modalities, such as PPG and ECG. Figure 1(b) illustrates knowledge distillation, which seeks to train a student model for the new modality to mimic the output of a pre-trained teacher model from the old modality (Abbaspourazad et al., 2024b; Zhang et al., 2024). The distillation process is memory intensive, as it requires forward inference with both the student and teacher models, in addition to backpropagation with the student model. The computational burden is further exacerbated by the emergence of large-scale biosignal foundation models (Coppola et al., 2024; Jiang et al., 2024; Pillai et al., 2025), which are mostly trained on specific modalities and have demonstrated exceptional performance across a wide range of tasks. Although these models offer tremendous performance gains, their use in cross-modal knowledge transfer is hindered by their size, which makes traditional knowledge distillation solutions computationally prohibitive for users without access to high-end GPUs. For example, distilling knowledge from PaPaGei, a PPG-based foundation model (Pillai et al., 2025), to the ECG-FM student model (McKeen et al., 2024) on the WESAD dataset (Schmidt et al., 2018), with a batch size of eight, requires more than 32GB of VRAM. Furthermore, because of data sharing regulations and privacy concerns, such a distillation process often needs to be performed locally where the data resides, under low-resource conditions. These constraints call for the development of an effective and efficient cross-modal transfer framework that can fully leverage the representation capability and embedded knowledge of the foundation models.

To this end, we propose BioX-Bridge, a new framework for unsupervised cross-modal knowledge transfer via model bridging, as illustrated in Figure 1(c). The core idea is to construct a bridge that projects intermediate representations from one biosignal model to another, leveraging the powerful representational capability and the rich embedded knowledge of foundation models.[1] The framework comprises two key components: bridge position selection and bridge architecture design. Specifically, we introduce an efficient two-stage strategy for selecting optimal input and output positions by evaluating the quality and similarity of intermediate representations between two biosignal models. To enable effective projection between high-dimensional spaces, we design a prototype network composed of a learnable prototype set and a low-rank approximation module to compute aggregation weights. Notably, only the bridge network requires training to enable interoperability between models of different modalities. We evaluate the effectiveness of BioX-Bridge in three biosignal datasets involving different modalities, demonstrating superior efficiency compared to existing methods. Extensive ablation studies further confirm the robustness of the proposed framework under various conditions. Our contributions can be summarized as follows:

---

[1]Note that the proposed BioX-Bridge framework is compatible with any deep learning-based biosignal models. In this work, we focus primarily on biosignal foundation models as a backbone to better support ongoing research in this area. Please refer to the appendix for ablation studies using different backbones.

- We propose BioX-Bridge, a novel unsupervised model bridging framework that enables cross-modal knowledge transfer through information flow between biosignal models.

- We introduce key components to support the framework, including an efficient two-stage strategy for selecting bridge positions and a prototype network with low-rank approximation for effective high-dimensional projection.

- We demonstrate the efficiency of BioX-Bridge through experiments on three biosignal datasets, four modalities, and six transfer directions, demonstrating robustness through comprehensive ablation studies.

## 2 RELATED WORKS

**Unsupervised Cross-modal Knowledge Transfer**   Existing methods can be divided into two categories: knowledge distillation and data translation.

*Knowledge distillation* was introduced as a model compression technique, where a smaller student model learns to mimic a larger and high-performing teacher model by matching its output distributions (Hinton et al., 2015). The concept has since been extended to cross-modal knowledge transfer. Early efforts focused on computer vision applications across a variety of sensor modalities, such as vision to depth images (Garcia et al., 2018; Gupta et al., 2016; Hoffman et al., 2016; Tian et al., 2020), to radio frequency heatmaps (Zhao et al., 2018), and to sound (Aytar et al., 2016; Xue et al., 2021). The core idea is to leverage unlabeled but semantically aligned data pairs to bridge the modality gap and transfer relevant knowledge to the corresponding tasks (Gou et al., 2021; Moslemi et al., 2024). Recent efforts have also investigated cross-modal knowledge distillation for biosignals. For example, Brant-X (Zhang et al., 2024) introduced a unified biosignal alignment framework that transfers knowledge from EEG to other biosignal modalities through a two-level semantic alignment strategy, such that the student model can provide complementary representations to the teacher model and improve downstream task performance. In another work (Abbaspourazad et al., 2024b), the distillation of knowledge from PPG to accelerometer signals was used to accurately predict physiological states such as heart rate. However, the aforementioned methods require training a full-size student model from scratch, which becomes increasingly impractical as model sizes grow, especially for resource-constrained settings.

*Data translation* aims to achieve unsupervised cross-modal knowledge transfer by directly translating raw data from one modality to another. Generative adversarial networks (GAN) (Goodfellow et al., 2020) and their variants (Mirza & Osindero, 2014; Zhu et al., 2017) have been widely adopted for modality translation tasks in the visual and signal processing domains (Duan et al., 2021; Sikka et al., 2021; Yang et al., 2020). A recent work (Wang et al., 2023) leveraged knowledge graphs to learn transformations between independently trained foundation models for proteins, drugs, and text. Nevertheless, reliance on structured knowledge graphs limits their applicability to biosignal scenarios, where such structured relationships are scarce or nonexistent. In the biosignal domain, cross-modal translation efforts have been largely limited to translation from PPG to ECG (Sarkar & Etemad, 2021; Zhu et al., 2021). Extending such translation to other modalities, such as EEG to ECG, remains largely underexplored.

**Biosignals Foundation Models**   Inspired by the recent success of large-scale pre-training in natural language processing (Achiam et al., 2023) and computer vision (Dosovitskiy et al., 2020), the development of foundation models for biosignals has garnered much interest (Han et al., 2024; Lai et al., 2025). Through large-scale self-supervised training on public and private biosignal datasets, several biosignal foundation models have been developed to capture rich and transferable representations, enabling more robust and efficient downstream adaptation. These models span a variety of modalities, including EEG (Chen et al., 2025a; 2024; Cui et al., 2023; Jiang et al., 2024; Wang et al., 2024), ECG (Coppola et al., 2024; Li et al., 2024; McKeen et al., 2024), PPG (Chen et al., 2025b; Pillai et al., 2025; Saha et al., 2025), accelerometer (Abbaspourazad et al., 2024b), and general-purpose biosignal models (Yang et al., 2023). In addition to unimodal models, recent work has explored multimodal foundation models for various applications such as health monitoring (Abbaspourazad et al., 2024a; Luo et al., 2024), sleep (Thapa et al., 2024), and activity recognition (Narayanswamy et al., 2024). Despite their strong performance on data from modalities seen during

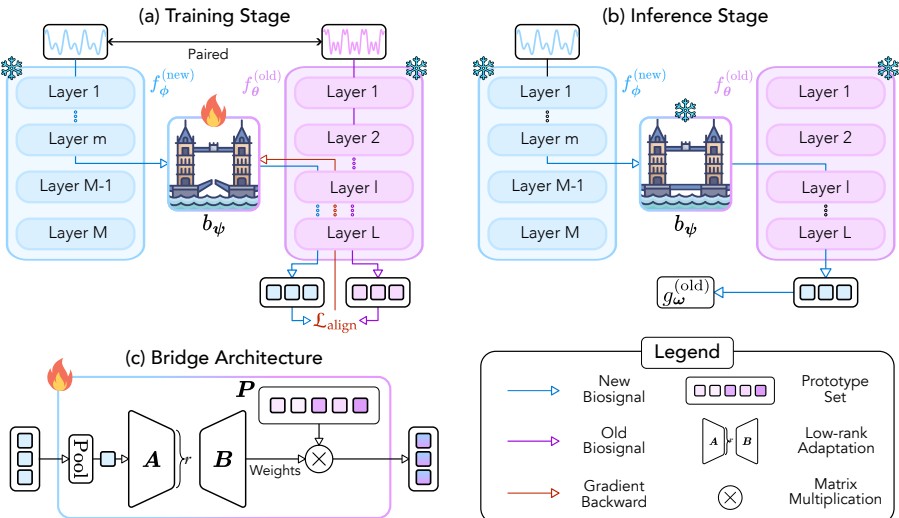

Figure 2: Overview of BioX-Bridge. (a) At the training stage, the bridge learns to project intermediate representations from the new modality to the old modality, such that it mimics the output of the old modality model. (b) At the inference stage, the bridge has been constructed and enables the flow of information between the two models in order to make predictions on data from the new modality. (c) The bridge consists of a low-rank approximation module and a prototype set. The low-rank approximation module generates aggregation weights for the prototype vectors.

pre-training, these models struggle with generalization to unseen modalities due to mismatches in input dimensions and data distributions (Liu et al., 2024).

We provide further discussion on model stitching and domain adaptation in Appendix C.

## 3 METHODS

The core concept of our proposed BioX-Bridge framework is to build a bridging network that facilitates efficient and effective projection between intermediate representations of biosignal models. This allows the framework to harness the strong representational power of one model while integrating the task-specific knowledge contained in another. We define the problem in Section 3.1 and introduce the idea of model bridging in Section 3.2. We detail the position of the bridge, its architecture and training in Sections 3.3–3.5. An overview of BioX-Bridge is presented in Figure 2.

### 3.1 PROBLEM DEFINITION

Assume that we are given an annotated dataset from an old biosignal modality, $\mathcal{D}^{(\mathrm{old})} = \{(\boldsymbol{x}_{i'}^{(\mathrm{old})}, y_{i'}^{(\mathrm{old})})\}_{i'=1}^{|\mathcal{D}^{(\mathrm{old})}|}$ with $|\mathcal{D}^{(\mathrm{old})}|$ labeled samples for a specific task, and a corresponding model, $g_{\boldsymbol{\omega}}^{(\mathrm{old})} \circ f_{\boldsymbol{\theta}}^{(\mathrm{old})}$, where $f_{\boldsymbol{\theta}}^{(\mathrm{old})}$ is a pre-trained encoder parametrized by $\boldsymbol{\theta}$ followed by a task head $g_{\boldsymbol{\omega}}^{(\mathrm{old})}$ parametrized by $\boldsymbol{\omega}$. We also have an un-annotated dataset from a new modality, $\mathcal{D}^{(\mathrm{new})} = \{\boldsymbol{x}_{i'}^{(\mathrm{new})}\}_{i'=1}^{|\mathcal{D}^{(\mathrm{new})}|}$, which shares the same underlying label set with $\mathcal{D}^{(\mathrm{old})}$. We further have a disjoint, un-annotated paired dataset $\mathcal{D}^{(\mathrm{pair})} = \{(\boldsymbol{x}_i^{(\mathrm{old})}, \boldsymbol{x}_i^{(\mathrm{new})})\}_{i=1}^{|\mathcal{D}^{(\mathrm{pair})}|}$. The unsupervised cross-modal knowledge transfer problem aims to obtain a model $f$, such that $f$ can make predictions on $\mathcal{D}^{(\mathrm{new})}$.

### 3.2 MODEL BRIDGING

Let $f_{\boldsymbol{\theta}}^{(\mathrm{old})}$ be the model for the old modality, parametrized by $\boldsymbol{\theta}$ of $L$ layers, and let $f_{\boldsymbol{\phi}}^{(\mathrm{new})}$ be the model for the new modality, parametrized by $\boldsymbol{\phi}$ of $M$ layers. The intermediate representations from the $m$-th layer of the new modality model can then be extracted as:

$$\boldsymbol{h}_m^{(\mathrm{new})} = f_{\boldsymbol{\phi}_{\leq m}}^{(\mathrm{new})}\left(\boldsymbol{x}^{(\mathrm{new})}\right), \qquad (1)$$

where $\boldsymbol{x}^{(\text{new})}$ denotes a biosignal time series sample from the new modality. $f_{\boldsymbol{\phi}_{\leq m}}^{(\text{new})}$ denotes the subset of the new modality model consisting of its first $m$ layers, subject to the constraint $1 \leq m \leq M$. $\boldsymbol{h}_m^{(\text{new})} \in \mathbb{R}^{N_m^{(\text{new})} \times d_m^{(\text{new})}}$ denotes the intermediate representation from the $m$-th layer of the new modality model with $N_m^{(\text{new})}$ number of tokens for transformer or spatial dimension for CNN and $d_m^{(\text{new})}$ token embedding dimension for transformer or number of channels for CNN.

Next, we introduce a bridge network to enable the information flow between the new and old modality models by projecting representations from the new modality into the representation space of the old modality:

$$\tilde{\boldsymbol{h}}_l^{(\text{old})} = b_{\boldsymbol{\psi}} \left( \boldsymbol{h}_m^{(\text{new})} \right), \tag{2}$$

where $b_{\boldsymbol{\psi}}$ denotes the bridge network parametrized by $\boldsymbol{\psi}$. $\tilde{\boldsymbol{h}}_l^{(\text{old})}$ denotes the projected representation from the new modality to the old modality. Note that the projected representation is designed to mimic the intermediate representation from the $l$-th layer of the old modality model, defined as $\boldsymbol{h}_l^{(\text{old})} = f_{\boldsymbol{\theta}_{\leq l}}^{(\text{old})} \left( \boldsymbol{x}^{(\text{old})} \right)$, where $\boldsymbol{x}^{(\text{old})}$ is the paired input signal from the old modality. Thus, $\tilde{\boldsymbol{h}}_l^{(\text{old})}, \boldsymbol{h}_l^{(\text{old})} \in \mathbb{R}^{N_l^{(\text{old})} \times d_l^{(\text{old})}}$ are of the same dimension.

Finally, we can obtain predictions using the old modality model starting from the $(l+1)$-th layer:

$$\tilde{y} = g_{\boldsymbol{\omega}}^{(\text{old})} \circ f_{\boldsymbol{\theta}_{>l}}^{(\text{old})} \left( \tilde{\boldsymbol{h}}_l^{(\text{old})} \right) = g_{\boldsymbol{\omega}}^{(\text{old})} \circ f_{\boldsymbol{\theta}_{>l}}^{(\text{old})} \circ b_{\boldsymbol{\psi}} \circ f_{\boldsymbol{\phi}_{\leq m}}^{(\text{new})} \left( \boldsymbol{x}^{(\text{new})} \right), \tag{3}$$

where $m$ and $l$ are also known as the bridge input and output positions, $\circ$ denotes function composition.

## 3.3 BRIDGE POSITION SELECTION

There are $L \times M$ possible locations where the bridge can be constructed between the layers of the two models. Although a brute-force search would yield the optimal bridge position, it is computationally expensive. In particular, the choice of the bridge position is one of the most influential factors affecting transfer performance, as we will show in ablation studies. To this end, we propose a two-stage strategy for efficient bridge position selection, as illustrated in Figure 3.

**Stage 1: Bridge Input Position** $(m)$ **Selection**   The bridge serves to project new modality representations to the old modality representation space, enabling the bridged model to mimic the behavior of the old modality model. As the saying "garbage in, garbage out" suggests, it is important to select discriminative new modality representations that can effectively distinguish among the predictions produced by the old modality model, also known as pseudo labels. We propose to select the input position of the bridge by linear probing, which has been widely used to evaluate the quality of intermediate representations (Alain & Bengio, 2016). The bridge input position selection can be formulated as:

$$\underset{m \in \{1,\dots,M\}}{\arg\min} \frac{1}{|\mathcal{D}^{(\text{pair})}|} \sum_{i=1}^{|\mathcal{D}^{(\text{pair})}|} \mathcal{L}_{\text{probe}} \left( g_{\boldsymbol{\eta}} \left( \boldsymbol{h}_{m,i}^{(\text{new})} \right), \hat{y}_i \right), \tag{4}$$

where $\boldsymbol{h}_{m,i}^{(\text{new})} = f_{\boldsymbol{\phi}_{\leq m}}^{(\text{new})}(\mathbf{x}_i^{(\text{new})})$ denotes the $i$-th sample's intermediate representation from the $m$-th layer of the new modality model, and $\hat{y}_i = g_{\boldsymbol{\omega}}^{(\text{old})} \circ f_{\boldsymbol{\theta}}^{(\text{old})}(\mathbf{x}_i^{(\text{old})})$ denotes the pseudo label. $\mathcal{L}_{\text{probe}}$ denotes the empirical loss for the linear prober $g_{\boldsymbol{\eta}}$. Note that the pseudo labels are simply the argmax of the teacher logits used in traditional knowledge distillation methods.

**Stage 2: Bridge Output Position** $(l)$ **Selection**   Since $\tilde{\boldsymbol{h}}_l^{(\text{old})} = b_{\boldsymbol{\psi}} \left( \boldsymbol{h}_m^{(\text{new})} \right)$ is designed to mimic $\boldsymbol{h}_l^{(\text{old})}$, we can ease the transformation process by selecting $\boldsymbol{h}_l^{(\text{old})}$ to be as similar as possible to $\boldsymbol{h}_m^{(\text{new})}$. We select linear CKA (Kornblith et al., 2019) as a well-established measure to find correspondences between the intermediate representations of neural networks. Let $\boldsymbol{H}_m^{(\text{new})} \in \mathbb{R}^{|\mathcal{D}^{(\text{pair})}| \times N_m^{(\text{new})} d_m^{(\text{new})}}$ denote the matrix of new modality representations extracted from the $m$-th layer , where the $i$-th row corresponds to the flattened $\boldsymbol{h}_{m,i}^{(\text{new})}$. Similarly, let $\boldsymbol{H}_l^{(\text{old})} \in \mathbb{R}^{|\mathcal{D}^{(\text{pair})}| \times N_l^{(\text{old})} d_l^{(\text{old})}}$ denote the matrix of old modality representations from the $l$-th layer. The bridge output position selection can be formulated

as:

$$\underset{l \in \{1,\ldots,L\}}{\arg\max} \operatorname{CKA}_{\text{linear}} \left( \boldsymbol{H}_m^{(\text{new})}, \boldsymbol{H}_l^{(\text{old})} \right) \tag{5}$$

For detailed formulation of $\operatorname{CKA}_{\text{linear}}$, please refer to the appendix.

---

**Algorithm 1:** BioX-Bridge learning procedure

---

**Input:** Old modality model $f_{\boldsymbol{\theta}}^{(\text{old})}$;

New modality model $f_{\boldsymbol{\phi}}^{(\text{new})}$;

Task head $g_{\boldsymbol{\omega}}^{(\text{old})}$; Paired dataset $\mathcal{D}^{(\text{pair})}$

**Output:** BioX-Bridge $g_{\boldsymbol{\omega}}^{(\text{old})} \circ f_{\boldsymbol{\theta}_{>l}}^{(\text{old})} \circ b_{\boldsymbol{\psi}} \circ f_{\boldsymbol{\phi}_{\leq m}}^{(\text{new})} (\cdot)$

**Init:** Bridge network $b_{\boldsymbol{\psi}} = \{\boldsymbol{A}, \boldsymbol{B}, \boldsymbol{P}\}$

1 ▷ **Bridge Position Selection**
2 Select bridge input position, $m$, using Eq. (4)
3 Select bridge output position, $l$, using Eq. (5)
4 ▷ **Bridge Training**
5 **for** epoch $\leftarrow 1$ to $n_{\text{epoch}}$ **do**
6    **for** step $\leftarrow 1$ to $n_{\text{step}}$ **do**
7       Sample mini-batch
      $\{(\boldsymbol{x}_i^{(\text{old})}, \boldsymbol{x}_i^{(\text{new})})\}_{i=1}^{\text{bs}} \subset \mathcal{D}^{(\text{pair})}$
8       Compute $\boldsymbol{h}_{L,i}^{(\text{old})} = f_{\boldsymbol{\theta}}^{(\text{old})} \left( \boldsymbol{x}_i^{(\text{old})} \right)$
9       Compute $\tilde{\boldsymbol{h}}_{L,i}^{(\text{old})} = f_{\boldsymbol{\theta}_{>l}}^{(\text{old})} \circ b_{\boldsymbol{\psi}} \circ f_{\boldsymbol{\phi}_{\leq m}}^{(\text{new})} \left( \boldsymbol{x}_i^{(\text{new})} \right)$
10       Compute $\mathcal{L}_{\text{align}} \left( \boldsymbol{h}_{L,i}^{(\text{old})}, \tilde{\boldsymbol{h}}_{L,i}^{(\text{old})} \right)$ using Eq. (7)
11       Update $\boldsymbol{\psi}$ w.r.t. gradients using $\nabla_{\boldsymbol{\psi}} \mathcal{L}$

12 ▷ **Bridge Inference**
13 **for** step $\leftarrow 1$ to $n_{\text{step}}$ **do**
14    Sample mini-batch $\{\boldsymbol{x}_{i'}^{(\text{new})}\}_{i'=1}^{\text{bs}} \subset \mathcal{D}^{(\text{new})}$
15    $\tilde{y}_{i'} = g_{\boldsymbol{\omega}}^{(\text{old})} \circ f_{\boldsymbol{\theta}_{>l}}^{(\text{old})} \circ b_{\boldsymbol{\psi}} \circ f_{\boldsymbol{\phi}_{\leq m}}^{(\text{new})} \left( \boldsymbol{x}_{i'}^{(\text{new})} \right)$

---

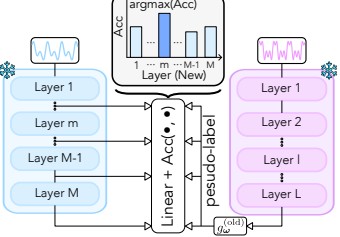

(a) Bridge Input Position Selection

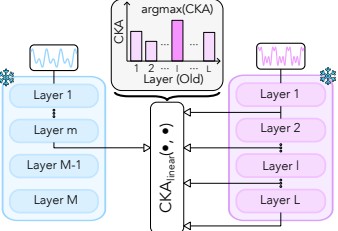

(b) Bridge Output Position Selection

Figure 3: Bridge Position Selection Strategy. For bridge input position, we select the layer from $f_{\boldsymbol{\phi}}^{(\text{new})}$ whose intermediate representation exhibits the strongest linear association with the pseudolabels. For bridge output position, we select the layer from $f_{\boldsymbol{\theta}}^{(\text{old})}$ whose representation is most similar to that of the bridge input layer.

### 3.4 BRIDGE ARCHITECTURE

Models of different modalities operate in distinct representational spaces. The bridge network should be sufficiently parametrized to enable the projection and alignment of the two spaces. A naive bridge architecture is a full-rank linear layer, but this is prohibitively expensive because of the high-dimensional projection from the new modality to the old modality. For example, using LaBraM (Jiang et al., 2024) as $f_{\boldsymbol{\phi}}^{(\text{new})}$ and HuBERT-ECG (Coppola et al., 2024) as $f_{\boldsymbol{\theta}}^{(\text{old})}$, the projection would require $N_m^{(\text{new})} \times d_m^{(\text{new})} \times N_l^{(\text{old})} \times d_l^{(\text{old})} = 181 \times 200 \times 93 \times 512 \approx 1.7$ billion parameters. To address the challenge of high-dimensional projection, we propose a prototype network. The prototype network consists of two modules, a prototype set, and a low-rank approximation module. The prototype set, $\boldsymbol{P} \in \mathbb{R}^{N_p \times d_l^{(\text{old})}}$, consisting of $N_p$ learnable prototype vectors with embedding dimension $d_l^{(\text{old})}$, introduces the flexibility to incorporate prior knowledge from $f_{\boldsymbol{\theta}}^{(\text{old})}$. To initialize the learnable prototype set, $\mathbf{P} \in \mathbb{R}^{N_p \times d_l^{(\text{old})}}$, we randomly select $N_p$ token/feature maps from $\mathbf{h}_l^{(\text{old})}$ of the training set samples. Specifically, $\mathbf{h}_l^{(\text{old})}$ are the intermediate representations from the $l$-th layer of the old modality model. The low-rank approximation module, consisting of $\boldsymbol{A} \in \mathbb{R}^{d_m^{(\text{new})} \times r}$ and $\boldsymbol{B} \in \mathbb{R}^{r \times N_l^{(\text{old})} N_p}$, reduces the number of trainable parameters through a low-rank factorization, while generating aggregation weights for prototype vectors, as illustrated in Figure 2.

$$\tilde{\boldsymbol{h}}_l^{(\text{old})} = \operatorname{Reshape}_{N_l^{(\text{old})} \times N_p} \left( \operatorname{Pool} \left( \boldsymbol{h}_m^{(\text{new})} \right) \otimes \boldsymbol{A} \otimes \boldsymbol{B} \right) \otimes \boldsymbol{P}, \tag{6}$$

where $\operatorname{Pool}(\cdot)$ denotes a pooling operation along the $N_m^{(\text{new})}$ dimension, and $\operatorname{Reshape}_{N_l^{(\text{old})} \times N_p}(\cdot)$ denotes the reshape operation to the specified output dimensions.

### 3.5 BRIDGE TRAINING

As the difference between $\boldsymbol{h}_l^{(\text{old})}$ and $\tilde{\boldsymbol{h}}_l^{(\text{old})}$ approaches zero, the bridged model yields predictions identical to those of the old modality model. Formally:

$$\boldsymbol{h}_l^{(\text{old})} = \tilde{\boldsymbol{h}}_l^{(\text{old})} \;\Rightarrow\; f_{\boldsymbol{\theta}_{>l}}^{(\text{old})}\left(\boldsymbol{h}_l^{(\text{old})}\right) = f_{\boldsymbol{\theta}_{>l}}^{(\text{old})}\left(\tilde{\boldsymbol{h}}_l^{(\text{old})}\right) \;\Rightarrow\; \boldsymbol{h}_L^{(\text{old})} = \tilde{\boldsymbol{h}}_L^{(\text{old})} \;\Rightarrow\; \hat{y} = \tilde{y}.$$

Naturally, the training objective for the bridge network is to align the intermediate representations in the $L$-th layer [2]:

$$\arg\min_{\boldsymbol{\psi}} \mathcal{L}_{\text{align}}\left(\boldsymbol{h}_L^{(\text{old})}, \tilde{\boldsymbol{h}}_L^{(\text{old})}\right) = \arg\min_{\boldsymbol{\psi}} \mathcal{L}_{\text{align}}\left(f_{\boldsymbol{\theta}}^{(\text{old})}\left(\boldsymbol{x}^{(\text{old})}\right), f_{\boldsymbol{\theta}_{>l}}^{(\text{old})} \circ b_{\boldsymbol{\psi}} \circ f_{\boldsymbol{\phi}_{\leq m}}^{(\text{new})}\left(\boldsymbol{x}^{(\text{new})}\right)\right), \quad (7)$$

where $\mathcal{L}_{\text{align}}$ denotes the loss function, such as cosine loss and mean absolute error loss. The learning process of BioX-Bridge is presented in Algorithm 1.

## 4 EXPERIMENTS

### 4.1 EXPERIMENTAL SETUPS

**Datasets & Tasks & Metrics** We consider the following datasets in the evaluation: (i) **WESAD** (Schmidt et al., 2018) is a wearable stress and affect detection dataset consisting of synchronized data from a wrist- and chest-worn device, collected from 15 subjects during a lab study. We select the ECG and PPG modality for a three-class (baseline/amusement/stress) classification task. (ii) **FOG** (Zhang et al., 2022) is a multimodal dataset for detecting freezing of gait in Parkinson's Disease, collected from 12 patients. We select the EMG and EEG modality for a two-class (normal/FOG) classification task. (iii) **ISRUC** (Khalighi et al., 2016) is a sleep-staging dataset consisting of synchronized data from a polysomnography, collected from 118 subjects in a laboratory study. We select the EEG and ECG modality for a two-class (sleep/wake) classification task. Dataset splits and preprocessing are detailed in the appendix. Given the unbalanced nature of the datasets, we report Balanced Accuracy, F1-Weighted, and F1-Macro, following (Jiang et al., 2024; Pillai et al., 2025).

**Backbone Foundation Models** For EEG, we adopt the base version of the LaBraM architecture with 5.8M parameters (Jiang et al., 2024). For ECG, we adopt the small version of the HuBERT-ECG architecture with 30.4M parameters (Coppola et al., 2024). For PPG, we adopt the small version of the PaPaGei architecture with 5.7M parameters (Pillai et al., 2025). For EMG, we adopt NormWear with 136.1M parameters (Luo et al., 2024). Note that LaBraM, HuBERT-ECG, and NormWear adopt a CNN-transformer architecture, while PaPaGei adopts a CNN architecture. All models are initialized with the pre-trained weights provided by the original publications. Note that biosignal foundation models are still early in their development, in comparison to foundation models for language and vision. Although current models contain a relatively small number of parameters, our method for efficient cross-modal knowledge transfer would be even more valuable as they scale up. As foundation models scale up, BioX-Bridge is efficient in terms of the bridge architecture to help reduce the number of trainable parameters. Furthermore, the bridge position selection strategy is lightweight in comparison to model training and scales linearly with the number of layers of the foundation models.

**Baselines** We compare our method with the following baselines evaluated on $\mathcal{D}^{(\text{new})}$: (i) Random denotes a model that produces predictions at random. (ii) CardioGAN uses GAN to synthesize ECG from PPG (Sarkar & Etemad, 2021), and we translate the new modality data (PPG) to the old modality (ECG) for evaluation. (iii) KD (Hinton et al., 2015) is the baseline of knowledge distillation. (iv) KD-contrast (Abbaspourazad et al., 2024b) is a variant of knowledge distillation with contrast loss (Zhang et al., 2024). (v) Oracle denotes the absolute best performance that can be achieved, which is simply the performance of the old modality model using old modality data. Please refer to the appendix for implementation details and further discussions.

---

[2]Note: While alignment can be performed at any layer between the $l$-th and $L$-th layers, we empirically find that performing alignment at the $L$-th (i.e., final) layer yields better transfer performance. We believe this is a result of error propagation. For alignment at the $l$-th layer, a small alignment error would grow as it propagates to the final layer. For alignment at the $L$-th layer, the error growth will be reflected in the alignment loss, enabling the bridge to take that into account and yield better downstream performance.

Table 1: Unsupervised cross-modal knowledge transfer performance on ISRUC, FOG, and WESAD. Oracle reflects supervised performance using the *old* modality only, also known as the teacher in knowledge distillation, whereas baselines and BioX-Bridge report performance on the *new* modality. Results are reported as mean across five seeds; standard deviations can be found in the appendix due to space constraints. The metrics are: Balanced Accuracy (BAcc), F1-M (F1-Macro), F1-W (F1-Weighted), Trainable Parameters (Params). Input indicates the data modality serving as the model's input during evaluation on the test set. The best unsupervised result is indicated in bold.

| **ISRUC** | | EEG (Old) → ECG (New) | | | | ECG (Old) → EEG (New) | | | | |
|---|---|---|---|---|---|---|---|---|---|---|
| **Methods** | Input | BAcc ↑ | F1-M ↑ | F1-W ↑ | Params ↓ | Input | BAcc ↑ | F1-M ↑ | F1-W ↑ | Params ↓ |
| Random | - | 50.00 | 46.48 | 53.52 | - | - | 50.00 | 46.48 | 53.52 | - |
| KD | ECG | 60.24 | 61.01 | 72.96 | 30.4M | EEG | 62.24 | 63.69 | 75.27 | 5.8M |
| KD-Contrast | ECG | **60.66** | 56.56 | 63.57 | 30.4M | EEG | **65.92** | 62.91 | 70.27 | 5.8M |
| BioX-Bridge | ECG | 60.11 | **61.20** | **74.02** | **1.8M** | EEG | 62.55 | **64.37** | **76.42** | **0.2M** |
| Oracle (Supervised) | EEG | 80.13 | 82.06 | 87.19 | - | ECG | 63.54 | 65.54 | 76.86 | - |

| **FOG** | | EEG (Old) → EMG (New) | | | | EMG (Old) → EEG (New) | | | | |
|---|---|---|---|---|---|---|---|---|---|---|
| **Methods** | Input | BAcc ↑ | F1-M ↑ | F1-W ↑ | Params ↓ | Input | BAcc ↑ | F1-M ↑ | F1-W ↑ | Params ↓ |
| Random | - | 50.00 | 49.99 | 50.01 | - | - | 50.00 | 49.99 | 50.01 | - |
| KD | EMG | 68.64 | 67.62 | 67.78 | 136.1M | EEG | 68.03 | 67.73 | 67.75 | 5.8M |
| KD-Contrast | EMG | 72.21 | 71.95 | 71.95 | 136.1M | EEG | **68.51** | 67.95 | 67.90 | 5.8M |
| BioX-Bridge | EMG | **72.24** | **72.12** | **72.16** | **1.2M** | EEG | 68.04 | **68.22** | **68.24** | **0.7M** |
| Oracle (Supervised) | EEG | 72.15 | 72.14 | 72.20 | - | EMG | 87.55 | 87.58 | 87.60 | - |

| **WESAD** | | ECG (Old) → PPG (New) | | | | PPG (Old) → ECG (New) | | | | |
|---|---|---|---|---|---|---|---|---|---|---|
| **Methods** | Input | BAcc ↑ | F1-M ↑ | F1-W ↑ | Params ↓ | Input | BAcc ↑ | F1-M ↑ | F1-W ↑ | Params ↓ |
| Random | - | 33.33 | 31.29 | 35.38 | - | - | 33.33 | 31.29 | 35.38 | - |
| CardioGAN | PPG | 39.32 | 19.63 | 20.33 | 28.2M | - | - | - | - | - |
| KD | PPG | 47.86 | **43.08** | 45.75 | 5.7M | ECG | 47.03 | 46.36 | 60.29 | 30.4M |
| KD-Contrast | PPG | 45.31 | 42.75 | 47.20 | 5.7M | ECG | 50.85 | 49.31 | 63.72 | 30.4M |
| BioX-Bridge | PPG | **49.57** | 42.28 | **47.44** | **0.2M** | ECG | **52.02** | **52.62** | **65.12** | **0.4M** |
| Oracle (Supervised) | ECG | 49.47 | 51.05 | 62.48 | - | PPG | 62.96 | 60.97 | 74.52 | - |

## 4.2 Unsupervised Cross-Modal Knowledge Transfer Performance

Experiment results on the ISRUC, FOG, and WESAD dataset are presented in Table 1. We observe that BioX-Bridge significantly reduces the number of trainable parameters by 87.9-99.1% and continues to achieve performance comparable to or better than that of the baseline methods. For example, for WESAD (PPG → ECG), BioX-Bridge requires merely 1.3% of trainable parameters while outperforming the baseline methods by around 1–2% across all metrics.

We also observe that the knowledge transfer performance gap compared to the oracles varies between datasets and knowledge transfer directions. For example, on the ISRUC dataset, we observe approximately 20% balanced accuracy gap between BioX-Bridge (60.11%) and the supervised EEG oracle (80.13%) for EEG → ECG, but only 1% gap between BioX-Bridge (62.55%) and the supervised ECG oracle (63.54%) for ECG → EEG. From this, we can draw a few observations: (1) Based on oracle results, the ISRUC task is more difficult for ECG (63.54%) than for EEG (80.13%), which is reasonable, as EEG has been considered the gold standard for sleep. (2) Our unsupervised training with BioX-Bridge can effectively produce an ECG model (60.11%) that performs similarly to the supervised ECG oracle (63.54%), despite the absence of labeled data. (3) There is a 20% gap between BioX-Bridge (60.11%) and the supervised EEG Oracle (80.13%) because the baselines and BioX-Bridge use the ECG as input, which is a less physiologically relevant modality for sleep than the EEG used by the supervised EEG oracle. (4) Unsupervised cross-modal knowledge transfer performance using EEG (62.55%) is also constrained by the performance of the ECG teacher (63.54%), a consequence inherent to knowledge transfer itself. Therefore, our unsupervised training produces a

relatively weaker EEG model (62.55%) compared to supervised EEG training (80.13%) since we are transferring from a weaker ECG model (63.54%).

On another note, BioX-Bridge and KD-Contrast achieved higher balanced accuracy than Oracle on several occasions, respectively. This is possible because balanced accuracy is simply an average of recall across classes. Higher balanced accuracy scores and lower F1 scores reflect that knowledge transfer methods achieved better recall but worse precision than Oracle.

## 4.3 ABLATION STUDIES

We conduct ablation studies on the WESAD dataset and the direction of knowledge transfer (PPG $\rightarrow$ ECG). Additional results are presented in Appendix D.6.

**Bridge Rank and Prototype Set** We study the impact of different hyperparameters for the prototype network in Figures 4a and 4b. A performance drop is observed when the approximation rank and prototype set size are too small or too large, likely due to under-/over-parameterization of the bridge network. In particular, the performance peaks at around 0.75M parameters in both cases.

**Dataset Size** We reduce the size of the paired dataset for bridge training. We observe in Figure 4c that the transfer performance slowly decays by around 2% at 20% dataset size, showcasing the robustness of the bridge under the low-data regime.

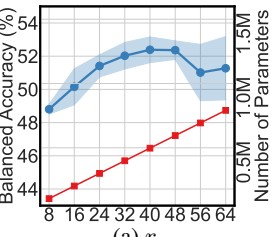 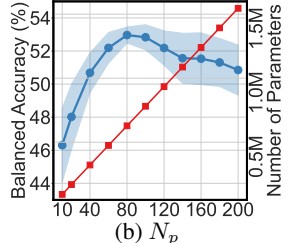 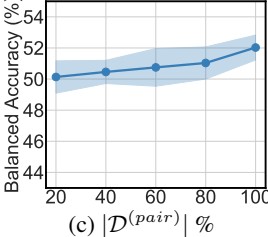

(a) $r$  (b) $N_p$  (c) $|\mathcal{D}^{(pair)}|$ %

Figure 4: Bridge Training Ablation. Blue: Balanced Accuracy. Red: Number of Parameters. We vary (a) bridge rank, (b) number of prototypes, and (c) pair dataset size to understand the robustness of BioX-Bridge and its performance under a low-data regime.

**Bridge Position Selection** To show that the bridge position selection strategy proposed in Section 3.3 is effective, we compare the unsupervised performance of cross-modal knowledge transfer at various positions in Table 2. For "Fixed", we train nine bridges in predefined positions, which is a combination of the first, middle, and last layers of the old and new modality models. The results are the average over all nine positions in five seeds. A breakdown of the result at each of the nine predefined positions is available in Appendix D.5.

**Foundation Model** We further analyze the impact of using different foundation models for cross-modal knowledge transfer. In Table 3, we replace the HuBERT-ECG foundation model (Coppola et al., 2024) with ECG-FM (McKeen et al., 2024). Notably, due to the large number of trainable parameters (90M), the knowledge distillation methods with ECG-FM could only be performed with a batch size of 4 on a V100 GPU. As a result, training for more than 50 epochs requires 6.5 hours for knowledge distillation methods and 1.9 hours for BioX-Bridge. Moreover, the performance gap between knowledge distillation methods and BioX-Bridge is much pronounced at 10–17%.

Table 2: Bridge Position Ablation. Comparison of different bridge position selection strategies with respect to BioX-Bridge. "Fixed" represents the average of 9 predefined positions, combining the first, middle, and last layers for both the input and output positions.

| Methods | BAcc ↑ | F1-M ↑ | F1-W ↑ |
|---|---|---|---|
| Fixed | 48.34 | 46.83 | 58.37 |
| BioX-Bridge | 52.02 | 52.62 | 65.12 |

Table 3: Foundation Model Ablation. We compare the transfer performance by replacing the ECG foundation model HuBERT-ECG with ECG-FM.

| Methods | Input | BAcc ↑ | F1-M ↑ | F1-W ↑ | Params ↓ |
|---|---|---|---|---|---|
| Random | - | 33.33 | 31.29 | 35.38 | - |
| KD | ECG | 48.44 | 45.84 | 54.18 | 90.8M |
| KD-Contrast | ECG | 43.06 | 42.94 | 54.21 | 90.8M |
| BioX-Bridge | ECG | **58.80** | **57.11** | **72.12** | **0.11M** |
| Oracle | PPG | 62.96 | 60.97 | 74.52 | - |

## 5 CONCLUSION

We present BioX-Bridge as an efficient framework for unsupervised cross-modal knowledge transfer across biosignals. To address the challenges of high-dimensional projection between biosignal foundation models, we design a prototype-based architecture for parameter-efficient learning of transformations between representation spaces. Our proposed two-stage bridge position selection strategy further identifies connection points that enable more effective alignment of intermediate representations. Through extensive experiments on diverse biosignal datasets and tasks, we demonstrated that BioX-Bridge achieves performance comparable to or superior to that of state-of-the-art methods while drastically reducing the number of trainable parameters. This work highlights the potential of model bridging as a powerful alternative to conventional cross-modal knowledge transfer techniques, offering a pathway to more accessible, adaptable, modality-agnostic, and resource-efficient biosignal applications in real-world settings, where computing resources and labelled data are often limited.

### ETHICAL AND REPRODUCIBILITY STATEMENT

This study makes use of datasets involving human subjects (ISRUC, WESAD, and FOG). All datasets employed are publicly available, and we follow the usage terms and ethical guidelines specified by the original data providers. No new data were collected for this work, and all analyses were conducted on de-identified, previously published datasets.

To ensure reproducibility, we provide detailed descriptions of our experimental setups, including data preprocessing steps, model architectures, hyperparameters, and training procedures, in Section 4.1 and Appendix D. Our code is available at: `https://github.com/chenqi-li/BioX-Bridge`.

### ACKNOWLEDGEMENTS

The work of Chenqi Li was supported by the Cyril and Phillis Long Scholarship at The Queen's College in partnership with the Clarendon Fund. The work of Tim Denison was funded by the Royal Academy of Engineering and supported by the NIHR Oxford Health Biomedical Research Centre (NIHR203316). The views expressed are those of the author(s) and not necessarily those of the NIHR or the Department of Health and Social Care. The Royal Academy of Engineering supported the work of Tingting Zhu under the Research Fellowship scheme.

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

APPENDIX

TABLE OF CONTENTS

## A   METHOD DETAILS: LINEAR CKA

Let $\boldsymbol{H}_m^{(\text{new})} \in \mathbb{R}^{|\mathcal{D}^{(\text{pair})}| \times N_m^{(\text{new})} d_m^{(\text{new})}}$ denote the matrix of new modality representations extracted from the $m$-th layer , where the $i$-th row corresponds to the flattened $\boldsymbol{h}_{m,i}^{(\text{new})}$. Similarly, let $\boldsymbol{H}_l^{(\text{old})} \in \mathbb{R}^{|\mathcal{D}^{(\text{pair})}| \times N_l^{(\text{old})} d_l^{(\text{old})}}$ denote the matrix of old modality representations from the $l$-th layer. The $\text{CKA}_{\text{linear}}$ operator introduced in Eq. 5 is formulated as follows (Kornblith et al., 2019):

$$\text{CKA}_{\text{linear}}(\boldsymbol{H}_m^{(\text{new})}, \boldsymbol{H}_l^{(\text{old})}) = \frac{\text{HSIC}(\boldsymbol{H}_m^{(\text{new})}, \boldsymbol{H}_l^{(\text{old})})}{\sqrt{\text{HSIC}(\boldsymbol{H}_m^{(\text{new})}, \boldsymbol{H}_m^{(\text{new})}) \cdot \text{HSIC}(\boldsymbol{H}_l^{(\text{old})}, \boldsymbol{H}_l^{(\text{old})})}},$$

$$\text{where} \quad \text{HSIC}(\boldsymbol{H}_m^{(\text{new})}, \boldsymbol{H}_l^{(\text{old})}) = \frac{1}{|\mathcal{D}^{(\text{pair})}|^2} \ \text{trace}\left(\boldsymbol{K}_t \boldsymbol{H} \boldsymbol{K}_s \boldsymbol{H}\right)$$

$$\text{and} \quad \boldsymbol{K}_t = \boldsymbol{H}_m^{(\text{new})}\left(\boldsymbol{H}_m^{(\text{new})}\right)^\top, \quad \boldsymbol{K}_s = \boldsymbol{H}_l^{(\text{old})}\left(\boldsymbol{H}_l^{(\text{old})}\right)^\top, \quad \boldsymbol{H} = \boldsymbol{I} - \frac{1}{|\mathcal{D}^{(\text{pair})}|}\boldsymbol{1}\boldsymbol{1}^\top.$$

HSIC is the Hilbert-Schmidt Independence Criterion. $\boldsymbol{H}$ is the centering matrix. Note that while this formulation uses the entire $\boldsymbol{H}_m^{(\text{new})}$ and $\boldsymbol{H}_l^{(\text{old})}$ to compute similarity between representations of the old and new modalities, it is also possible to improve efficiency by computing similarity using only a subset of their rows.

## B    LIMITATIONS AND FUTURE WORK

Limitations: Although BioX-Bridge greatly reduces training computational requirements and improves the efficiency of cross-modal knowledge transfer, it depends on the availability of pre-trained models for each biosignal modality, an assumption that may not hold for emerging or underexplored biosignals. Furthermore, depending on the position of the bridge, the inference time of the bridged model could be longer.

Future Work: Whether the unsupervised cross-modal knowledge transfer method should be task-agnostic is an interesting next step to explore. A task-agnostic method would offer better generality, making it suitable for multi-task scenarios. However, such approaches can potentially limit task-specific performance compared to task-specific methods. Investigating other data-free and task-agnostic bridge position selection strategies would further improve the flexibility of BioX-Bridge for different application scenarios. In addition, existing unsupervised cross-modal knowledge transfer frameworks rely on the availability of paired data, which is more difficult to collect than unimodal data. Future work focusing on transfer using unpaired data would extend applications to any combination of modalities. Future works can also explore BioX-Bridge for datasets with more than two modalities and investigate how BioX-Bridge generalizes across datasets that share the same modalities, allowing a deeper understanding of modality interactions and dataset-specific adaptations (Zhang et al., 2022b; Liu et al., 2023; Fang et al., 2024).

## C    ADDITIONAL DISCUSSION ON RELATED WORKS AND BASELINES

### C.1    MODEL STITCHING

Model stitching combines subsets of layers from two or more models to create a hybrid one, which was initially introduced as a metric to compare intermediate representations across neural networks by examining their compatibility through network recombination (Bansal et al., 2021; Csiszárik et al., 2021; Lenc & Vedaldi, 2015; Moschella et al., 2023). Most of the work focused mainly on stitching adjacent layers from models of varying sizes within the same architecture family. More recently, it has been revisited as a general strategy for leveraging families of pretrained models to construct scalable neural networks that can accommodate diverse deployment constraints (He et al., 2024; Pan et al., 2023; Yang et al., 2022). For the stitching layer, a simple 1x1 convolution layer is often sufficient for stitching, under the assumption that the intermediate representations are similar and that the token counts (in transformers) or spatial dimensions (in convolutional neural networks, CNNs) are aligned. However, this assumption breaks down when models attempt to stitch across different modalities, where intermediate representations differ in both semantics and dimensionality.

### C.2    DOMAIN ADAPTATION AND GENERALIZATION

Domain adaptation methods are typically homogeneous, where they focus on adapting a single model from a source domain to a target domain (Wilson & Cook, 2020). Existing foundation models are unimodal and expect a fixed input shape, but biosignals are heterogeneous, whereby they contain different numbers of channels, are sampled at different frequencies, and are segmented with different window sizes. For example, ECG-FM only accepts 12-lead, 5-second ECG segments at a sampling frequency of 250Hz, and cannot work on 6-channel, 30-second EEG segments at a sampling frequency of 200Hz. This means that homogeneous unsupervised domain adaptation methods will fail to work in our setting.

Heterogeneous domain adaptation methods are more suited for our setup, where they acknowledge the difference in modalities between the source and target data. Heterogeneous unsupervised domain adaptation assumes access to a labelled source domain and an unlabelled target domain for knowledge transfer (Yang et al., 2025; Liu et al., 2020), but our setting assumes access to data from unlabelled source and target domains. Therefore, heterogeneous domain adaptation methods are not suitable for the unsupervised+paired scenario in this work.

### C.3 CARDIOGAN FOR ADDITIONAL MODALITIES

In Table 1, we present CardioGAN as a baseline for WESAD ECG → PPG. We omit CardioGAN as a baseline for the remaining knowledge transfer directions because the extension of data translation to other modalities is non-trivial. CardioGAN was originally designed for synthesizing ECG from PPG, both of which are single leads. However, biosignals such as EEG, EMG, and ECG have multiple channels, and extending the generation to multiple channels using multi-channel input is non-trivial. The authors of CardioGAN mention that generating multi-lead ECGs is one of the future works. Additionally, to the best of our knowledge, no prior study has shown any benchmarks for translating between EEG and ECG and vice versa.

## D EXPERIMENT DETAILS

### D.1 SETUP AND IMPLEMENTATION DETAILS

#### D.1.1 DATASET PREPROCESSING

Each foundation model specifies its preprocessing pipeline, and we follow these procedures accordingly. If a notch or bandpass filter has already been applied to the dataset, we skip that step during preprocessing.

**WESAD** In this dataset, ECG signals are sampled at 700Hz and PPG signals at 64Hz. For ECG, which uses HuBERT-ECG as its foundation model, we first downsample to 500Hz, apply a finite impulse response (FIR) bandpass filter between 0.05–47Hz, resample to 100Hz, and then perform channel-wise z-score normalization. For PPG, which uses PaPaGei as its foundation model, we upsample the signals to 125Hz, apply a 4th-order Chebyshev bandpass filter between 0.5–12Hz, and normalize using z-score normalization. Finally, all recordings are segmented into 60-second windows with a 5-second step size.

**FOG** In this dataset, both EEG and EMG signals were collected at 1000Hz, downsampled to 500Hz with a notch and bandpass filter already applied. For EEG, which uses LaBraM as its foundation model, we downsample to 200Hz and convert the unit to 0.1mV. For EMG, which uses NormWear as its foundation model, we downsample to 130Hz and normalize using z-score normalization. All recordings are segmented into 3-second windows with a sliding step size of 0.3 seconds.

**ISRUC** In this dataset, EEG signals are sampled at 200Hz with a notch and bandpass filter already applied, while PPG signals are also sampled at 200Hz with a notch filter applied. For EEG, which uses LaBraM as its foundation model, no resampling is required; we only convert the unit to 0.1mV. For ECG, which uses HuBERT-ECG as its foundation model, we upsample to 500Hz, apply a bandpass filter between 0.05–47Hz, resample to 100Hz, and normalize using z-score normalization.

#### D.1.2 DATASET SPLIT

Dataset split is summarized in Figure A1. The datasets contain synchronized data from the old and new modalities. We perform a subject-wise split for WESAD and ISRUC and sample-wise split for FOG (Zhang et al., 2022) to obtain four subsets $\mathcal{D}^{(\text{old})}, \mathcal{D}^{(\text{new})}, \mathcal{D}^{(\text{val})}$, and $\mathcal{D}^{(\text{pair})}$, at a ratio of 33%, 22%, 11%, and 33%, respectively. We use old modality data from $\mathcal{D}^{(\text{old})}$ to train the linear prober $g_{\omega}^{(\text{old})}$. New modality data from $\mathcal{D}^{(\text{new})}$ is used to evaluate bridge performance in an unseen set. All data from $\mathcal{D}^{(\text{pair})}$ and $\mathcal{D}^{(\text{val})}$ are used to train and help select hyperparameters.

#### D.1.3 BIOX-BRIDGE AND BASELINE IMPLEMENTATION DETAILS

**BioX-Bridge Implementation Details** To prepare $f_{\boldsymbol{\theta}}^{(\text{old})}$ and $g_{\omega}^{(\text{old})}$ for evaluation, we adapt the pre-trained foundation models for the classification tasks using $\mathcal{D}^{(\text{old})}$, we apply mean pooling to the last-layer representations and add a linear layer for classification. The weights of the foundation model are frozen, and the linear layer is trained for more than 50 epochs. We select a learning rate of 1e-4 for LaBraM, 1e-4 for PaPaGei, 1e-4 for NormWear, and 1e-5 for HuBERT-ECG, as suggested in the original publications. To select the input position of the bridge, we use logistic regression with L2 regularization as the linear prober. We select the layer with the best F1 macro score over five-fold cross-validation. We train the bridge over 50 epochs with cosine embedding loss on $\mathcal{D}^{(\text{pair})}$.

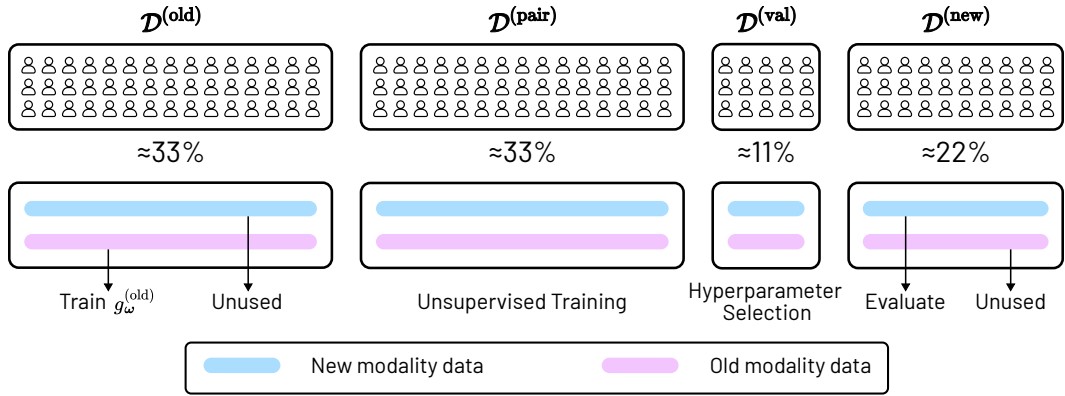

Figure A1: Illustration of Dataset Split. The dataset is divided into four subject-independent subsets. We first use the old modality data from $\mathcal{D}^{(\text{old})}$ to train the linear prober $g_\omega^{(\text{old})}$ for experiment setup, followed by unsupervised training on $\mathcal{D}^{(\text{pair})}$. The subsets $\mathcal{D}^{(\text{val})}$ and $\mathcal{D}^{(\text{new})}$ are used for hyperparameter selection and testing, respectively.

All experiments are conducted on V100 GPUs with 32GB VRAM. The training time for a single run ranges from 10 minutes to 4 hours, depending on the dataset and the backbone foundation model.

To select the bridge rank and number of prototypes, we perform a grid search over $r = \{4, 8, 16, 32\}$ and $N_p = \{50, 100, 150, 200, 250, 300\}$. For compatibility across various network architectures without modifying the forward functions, we retrieve and replace intermediate representations using forward hooks and forward pre-hooks [3]. Please see the source code for detailed implementation.

**Baseline Implementation Details** For the Random baseline, we simulate a model that randomly assigns labels to samples uniformly across all classes. For CardioGAN, we adopt the pre-trained weights provided by the original publication. We do not perform any fine-tuning as CardioGAN has been trained with WESAD as one of its datasets. We adopt the preprocessing code provided by CardioGAN to prepare the PPG. For each recording, we generate the corresponding ECG using a sliding window of 4 seconds and a step size of 4 seconds. The generated ECG recording is then preprocessed following the HuBERT-ECG pipeline described in Section D.1.1.

For knowledge distillation (KD), we append a linear layer to map mean-pooled representations from the last layer of the foundation model to classwise probabilities. We then fine-tune the entire foundation model with a learning rate of 1e-4 for LaBraM, 1e-4 for PaPaGei, 1e-4 for NormWear, and 1e-5 for HuBERT-ECG, as suggested in the original publications. For contrastive knowledge distillation (KD-Contrast), we append a linear layer to map mean-pooled representations from the last layer of the new modality foundation model to those of the old modality foundation model. During inference, we adopt the linear prober from the old modality foundation model to produce classwise probabilities. The temperature for the InfoNCE loss is selected as 0.04 following (Abbaspourazad et al., 2024b).

## D.2 Main Results with Standard Deviations

Tables 1–3 only report the average performance for five seeds due to space constraints. In Table A1–A5, we present the full results with standard deviation.

## D.3 Additional Ablation Studies

### D.3.1 BioX-Bridge Framework with Traditional Biosignal Models

All results presented in the main body focused on building a bridge between biosignal foundation models to better support ongoing research in this area. To show that BioX-Bridge is also compatible

---

[3]PyTorch documentation: forward hook, forward pre-hook

Table A1: Unsupervised cross-modal knowledge transfer performance on ISRUC. Results are reported as mean±std (%) across five seeds. Knowledge transfer direction is indicated as (old modality → new modality).

(a) **ISRUC** (EEG → ECG)

| Methods | Input Modality | Balanced Accuracy ↑ | F1 Macro ↑ | F1 Weighted ↑ | Trainable Parameters ↓ |
|---|---|---|---|---|---|
| Random | - | 50.00 | 46.48 | 53.52 | - |
| KD | ECG | 60.24±1.16 | 61.01±1.19 | 72.96±0.68 | 30.4M |
| KD-Contrast | ECG | **60.66±1.48** | 56.56±1.98 | 63.57±2.41 | 30.4M |
| BioX-Bridge | ECG | 60.11±1.17 | **61.20±1.39** | **74.02±0.95** | **1.8M** |
| Oracle | EEG | 80.13 | 82.06 | 87.19 | - |

(b) **ISRUC** (ECG → EEG)

| Methods | Input Modality | Balanced Accuracy ↑ | F1 Macro ↑ | F1 Weighted ↑ | Trainable Parameters ↓ |
|---|---|---|---|---|---|
| Random | - | 50.00 | 46.48 | 53.52 | - |
| KD | EEG | 62.24±1.82 | 63.69±2.05 | 75.27±1.26 | 5.8M |
| KD-Contrast | EEG | **65.92±0.94** | 62.91±1.60 | 70.27±1.95 | 5.8M |
| BioX-Bridge | EEG | 62.55±2.10 | **64.37±2.65** | **76.42±1.46** | **0.2M** |
| Oracle | ECG | 63.54 | 65.54 | 76.86 | - |

Table A2: Unsupervised cross-modal knowledge transfer performance on FOG. Results are reported as mean±std (%) across five seeds. Knowledge transfer direction is indicated as (old modality → new modality).

(a) **FOG** (EEG → EMG)

| Methods | Input Modality | Balanced Accuracy ↑ | F1 Macro ↑ | F1 Weighted ↑ | Trainable Parameters ↓ |
|---|---|---|---|---|---|
| Random | - | 50.00 | 49.99 | 50.01 | - |
| KD | EMG | 68.64±3.79 | 67.62±5.77 | 67.78±5.61 | 136.1M |
| KD-Contrast | EMG | 72.21±2.77 | 71.95±2.53 | 71.95±2.57 | 136.1M |
| BioX-Bridge | EMG | **72.24±0.63** | **72.12±0.60** | **72.16±0.60** | **1.2M** |
| Oracle | EEG | 72.15 | 72.14 | 72.20 | - |

(b) **FOG** (EMG → EEG)

| Methods | Input Modality | Balanced Accuracy ↑ | F1 Macro ↑ | F1 Weighted ↑ | Trainable Parameters ↓ |
|---|---|---|---|---|---|
| Random | - | 50.00 | 49.99 | 50.01 | - |
| KD | EEG | 68.03±2.22 | 67.73±2.63 | 67.75±2.71 | 5.8M |
| KD-Contrast | EEG | **68.51±2.08** | 67.95±1.98 | 67.90±2.01 | 5.8M |
| BioX-Bridge | EEG | 68.04±1.99 | **68.22±1.80** | **68.24±1.81** | **0.7M** |
| Oracle | EMG | 87.55 | 87.58 | 87.60 | - |

with traditional biosignal models, we replace HuBERT-ECG with a pre-trained ECG model, ECG-DualNet (Rohr et al., 2022), in Table A6. We notice that BioX-Bridge continues to outperform the baseline methods while significantly reducing the number of trainable parameters

Table A3: Unsupervised cross-modal knowledge transfer performance on WESAD. Results are reported as mean±std (%) across five seeds. Knowledge transfer direction is indicated as (old modality → new modality).

(a) **WESAD** (ECG → PPG)

| Methods | Input Modality | Balanced Accuracy ↑ | F1 Macro ↑ | F1 Weighted ↑ | Trainable Parameters ↓ |
|---|---|---|---|---|---|
| Random | - | 33.33 | 31.29 | 35.38 | - |
| CardioGAN | PPG | 39.32 | 19.63 | 20.33 | - |
| KD | PPG | 47.86±2.36 | **43.08±3.05** | 45.75±3.90 | 5.7M |
| KD-Contrast | PPG | 45.31±5.00 | 42.75±3.68 | 47.20±2.67 | 5.7M |
| BioX-Bridge | PPG | **49.57±5.51** | 42.28±3.22 | **47.44±9.18** | **0.2M** |
| Oracle | ECG | 49.47 | 51.05 | 62.48 | - |

(b) **WESAD** (PPG → ECG)

| Methods | Input Modality | Balanced Accuracy ↑ | F1 Macro ↑ | F1 Weighted ↑ | Trainable Parameters ↓ |
|---|---|---|---|---|---|
| Random | - | 33.33 | 31.29 | 35.38 | - |
| KD | ECG | 47.03±1.60 | 46.36±1.98 | 60.29±2.51 | 30.4M |
| KD-Contrast | ECG | 50.85±3.61 | 49.31±3.13 | 63.72±3.22 | 30.4M |
| BioX-Bridge | ECG | **52.02±0.80** | **52.62±0.36** | **65.12±0.91** | **0.4M** |
| Oracle | PPG | 62.96 | 60.97 | 74.52 | - |

Table A4: Bridge Position Ablation. WESAD (PPG → ECG). We study the effectiveness of the bridge selection strategy.

| Methods | Input Modality | Balanced Accuracy ↑ | F1 Macro ↑ | F1 Weighted ↑ |
|---|---|---|---|---|
| BioX-Bridge @ Fixed | ECG | 48.34±3.28 | 46.83±6.16 | 58.37±7.36 |
| BioX-Bridge @ Selected | ECG | 52.02±0.80 | 52.62±0.36 | 65.12±0.91 |

Table A5: Foundation Model Ablation. WESAD (PPG → ECG). We replace HuBERT-ECG with another ECG foundation model, ECG-FM.

| Methods | Input Modality | Balanced Accuracy ↑ | F1 Macro ↑ | F1 Weighted ↑ | Trainable Parameters ↓ |
|---|---|---|---|---|---|
| Random | - | 33.33 | 31.29 | 35.38 | - |
| KD | ECG | 48.44±8.07 | 45.84±8.80 | 54.18±13.46 | 90.8M |
| KD-Contrast | ECG | 43.06±4.33 | 42.94±3.88 | 54.21±8.81 | 90.8M |
| BioX-Bridge | ECG | **58.80±1.00** | **57.11±0.84** | **72.12±0.92** | **0.11M** |
| Oracle | PPG | 62.96 | 60.97 | 74.52 | - |

### D.3.2 IMPACT OF FOUNDATION MODEL SIZE

The HubERT-ECG family of models is available in three sizes, with varying numbers of parameters. We replace the small version of HuBERT-ECG (30M) with base (93M) and large (183M) versions, and report both baseline and our results in Table A7. We observe that the best transfer performance for BioX-Bridge is achieved using the large version of HuBERT-ECG, thanks to the more powerful

Table A6: Traditional Model Ablation. WESAD (PPG → ECG). We replace HuBERT-ECG with a traditional pre-trained model ECG-DualNet.

| Methods | Input Modality | Balanced Accuracy ↑ | F1 Macro ↑ | F1 Weighted ↑ | Trainable Parameters ↓ |
|---|---|---|---|---|---|
| Random | - | 33.33 | 31.29 | 35.38 | - |
| KD | ECG | 52.78±2.15 | 51.86±3.09 | 64.81±2.72 | 8.2M |
| KD-Contrast | ECG | 49.79±3.31 | 50.40±3.78 | 59.84±3.58 | 8.2M |
| BioX-Bridge | ECG | **56.10±1.21** | **53.25±1.54** | **67.60±1.92** | **0.4M** |
| Oracle | PPG | 62.96 | 60.97 | 74.52 | - |

representational capabilities. Overall, using base and large model yields better transfer performance than small model for KD and BioX-Bridge. The same is not true for KD-Contrast, which observes the best transfer performance using the small model, potentially as a result of overparametrization and limited training dataset for fine-tuning the larger models.

Table A7: Foundation Model Size Ablation. WESAD (PPG → ECG). We replace the small version of HuBERT-ECG (30M) with base (93M) and large (183M).

| Methods | Input Modality | Model Size | Balanced Accuracy ↑ | F1 Macro ↑ | F1 Weighted ↑ | Trainable Parameters ↓ |
|---|---|---|---|---|---|---|
| Random | - | - | 33.33 | 31.29 | 35.38 | - |
| KD | ECG | Small | 47.03±1.60 | 46.36±1.98 | 60.29±2.51 | 30.4M |
| KD | ECG | Base | 48.99±1.23 | 50.38±2.35 | 61.52±2.33 | 93.1M |
| KD | ECG | Large | 48.56±5.05 | 49.11±4.72 | 61.33±4.77 | 188.6M |
| KD-Contrast | ECG | Small | 50.85±3.61 | 49.31±3.13 | 63.72±3.22 | 30.4M |
| KD-Contrast | ECG | Base | 48.24±2.73 | 44.92±4.36 | 57.79±6.29 | 93.1M |
| KD-Contrast | ECG | Large | 49.31±3.88 | 48.55±4.95 | 62.18±5.80 | 188.6M |
| BioX-Bridge | ECG | Small | 52.02±0.80 | 52.62±0.36 | 65.12±0.91 | 0.4M |
| BioX-Bridge | ECG | Base | 52.46±0.97 | 50.44±2.29 | 64.69±2.97 | 0.5M |
| BioX-Bridge | ECG | Large | **54.94±1.77** | **53.23±0.93** | **67.90±0.75** | **0.8M** |
| Oracle | PPG | - | 62.96 | 60.97 | 74.52 | - |

## D.4 BRIDGE POSITION SELECTION RESULTS

In Figure A2–A4, we provide the layer-wise results for the bridge position selection strategy proposed in Section 3.3, including the pseudo label linear probing for bridge input position selection, and CKA similarity for bridge output position selection.

It is observed that the linear probing performance of pseudo labels using representations from different layer $m$ of $f_\phi^{(new)}$ varies quite significantly. The range for the linear probing performance across different layer $m$ is as low as 2% for FOG EEG → EMG and as high as 14% for WESAD ECG → PPG. Therefore, the difference in input intermediate representations across different layers varies between datasets and transfer directions. However, this difference is not solely a result of dataset characteristics, but also likely a result of the model architecture. In general, linear probing performance tends to increase as we retrieve representations from deeper layers of transformers (LaBraM, NormWear, HuBERT-ECG), and the same does not apply to CNN (PaPaGei), where we observed better linear probing performance in the shallower layers. A possible explanation for this difference lies in how the two architectures organize information across different layers. For CNN-based models, earlier layers capture fine-grained and low-level features, while the later layers capture coarse and high-level features with larger receptive fields. The lower-level features might

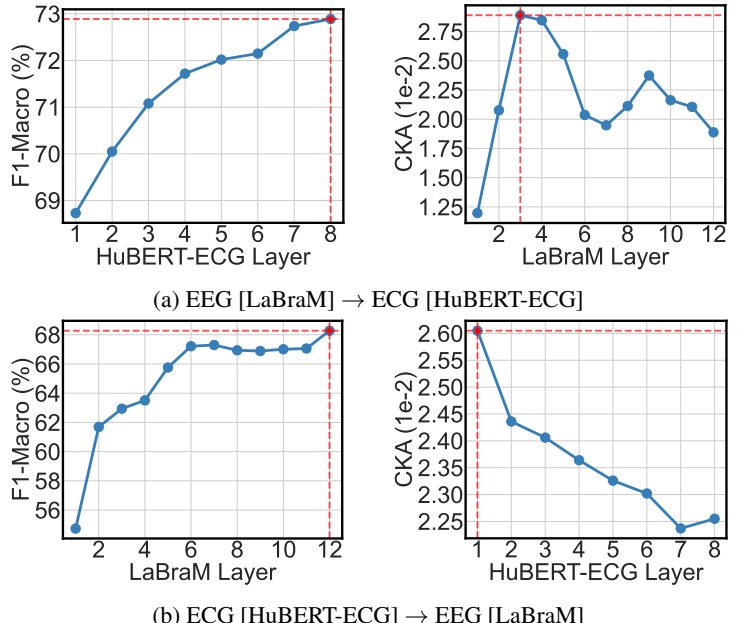

(a) EEG [LaBraM] → ECG [HuBERT-ECG]

(b) ECG [HuBERT-ECG] → EEG [LaBraM]

Figure A2: ISRUC bridge position selection. Knowledge transfer direction is indicated as (old modality $[f_\theta^{(\text{old})}]] \to$ new modality $[f_\phi^{(\text{new})}]$). The layer selected is highlighted using red dashed lines. (left) Bridge input position selection, where we select the layer from $f_\phi^{(\text{new})}$ whose intermediate representation exhibits the strongest linear association with the pseudolabels. (right) Bridge output position selection, where we select the layer from $f_\theta^{(\text{old})}$ whose representation is most similar to that of the selected bridge input layer.

be more informative for the task. Unlike CNN, transformer-based models have access to the same receptive field across all layers. As the model continues to refine and reorganize the representations across layers, the latter representation yields better linear probing performance.

For a given layer $m$ from $f_\phi^{(\text{new})}$, as we vary layer $l$ from $f_\theta^{(\text{old})}$, we do not observe a clear trend for the similarity of the representations between the two layers in the datasets and transfer directions studied. Note that the absolute CKA similarity, on the scale of 1e-2, is low across all datasets, where CKA ranges between 0 and 1. This is expected as the two models are trained on data from different modalities, and it is the relative difference in similarity between layer $l$ and a given layer $m$ that is of interest for bridge output position selection. Define the coefficient of variation (CV) for each transfer direction as:

$$\text{CV} = \frac{\sigma}{\mu} \times 100 = \frac{\sqrt{\frac{1}{L}\sum_{l=1}^{L}(c_l - \mu)^2}}{\frac{1}{L}\sum_{l=1}^{L} c_l} \times 100. \tag{8}$$

where $c_l$ is the CKA similarity between the representations of the layer $m$ of $f_\phi^{(\text{new})}$ and the layer $l$ of $f_\theta^{(\text{old})}$, and $\mu$ and $\sigma$ denote the mean and standard deviation of $\{c_l\}_{l=1}^{L}$, respectively. The CV is as low as 2.05 for WESAD ECG → PPG and as high as 30.72 for FOG EEG → EMG.

We further analyze the impact of the reduced dataset size on the bridge position selection results. Figure A5 shows the results of the bridge position selection in WESAD when using 50% of the data compared to Figure A4. Comparing these two figures, we observe very similar trends for the linear probing and CKA similarity plots. For PPG → ECG, the same bridge input and output layers are selected. For ECG → PPG, the bridge input layer differs by one layer while the bridge output layer is the same. This suggests that the bridge position selection strategy is relatively stable with respect to the dataset used.

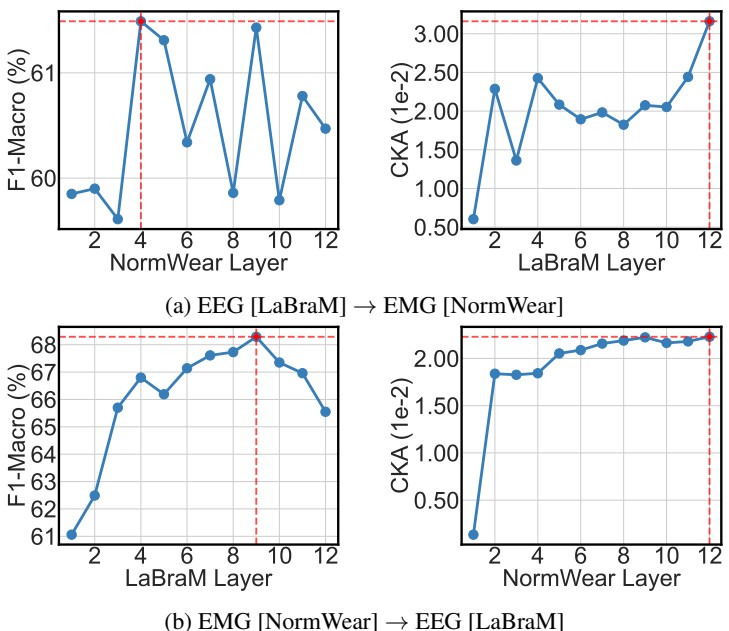

Figure A3: FOG bridge position selection. Knowledge transfer direction is indicated as (old modality $[f_\theta^{(old)}] \to$ new modality $[f_\phi^{(new)}]$). The layer selected is highlighted using red dashed lines. (left) Bridge input position selection, where we select the layer from $f_\phi^{(new)}$ whose intermediate representation exhibits the strongest linear association with the pseudolabels. (right) Bridge output position selection, where we select the layer from $f_\theta^{(old)}$ whose representation is most similar to that of the selected bridge input layer.

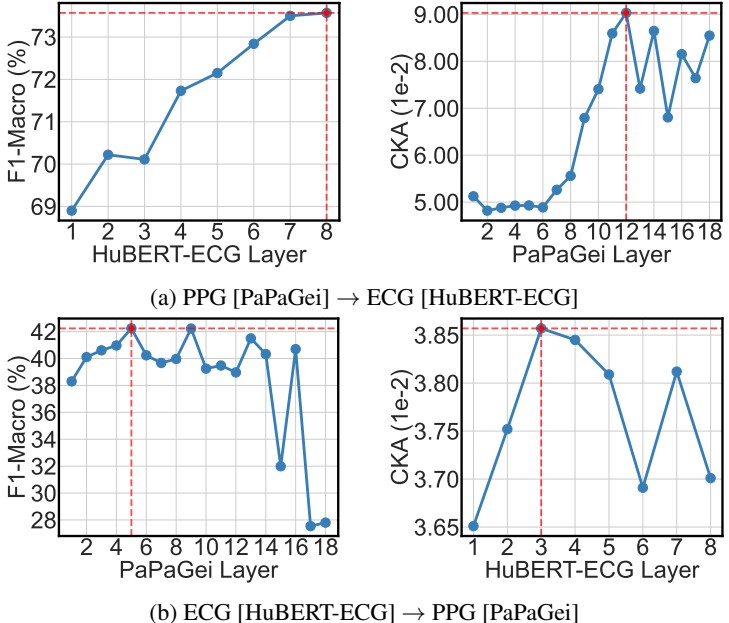

Figure A4: WESAD bridge position selection. Knowledge transfer direction is indicated as (old modality $[f_\theta^{(old)}] \to$ new modality $[f_\phi^{(new)}]$). The layer selected is highlighted using red dashed lines. (left) Bridge input position selection, where we select the layer from $f_\phi^{(new)}$ whose intermediate representation exhibits the strongest linear association with the pseudolabels. (right) Bridge output position selection, where we select the layer from $f_\theta^{(old)}$ whose representation is most similar to that of the selected bridge input layer.

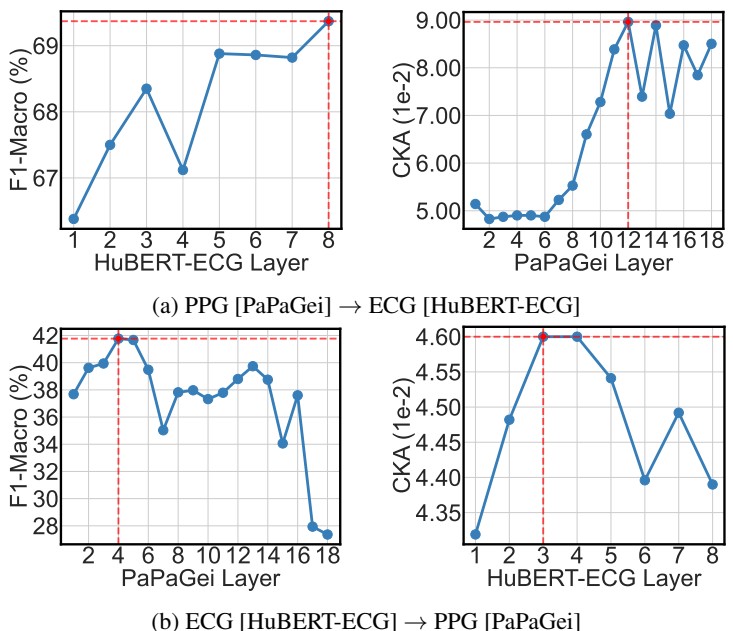

(a) PPG [PaPaGei] → ECG [HuBERT-ECG]

(b) ECG [HuBERT-ECG] → PPG [PaPaGei]

Figure A5: WESAD bridge position selection with 50% data. Knowledge transfer direction is indicated as (old modality $[f_\theta^{(old)}]$ → new modality $[f_\phi^{(new)}]$). The layer selected is highlighted using red dashed lines. (left) Bridge input position selection, where we select the layer from $f_\phi^{(new)}$ whose intermediate representation exhibits the strongest linear association with the pseudolabels. (right) Bridge output position selection, where we select the layer from $f_\theta^{(old)}$ whose representation is most similar to that of the selected bridge input layer.

### D.5 BRIDGE POSITION ABLATION RESULTS

To better understand the impact of bridge position selection, we provide a breakdown of the performance at each of the nine predefined positions from Table 2 in the form of heatmaps in Figure A6–A8.

In terms of the bridge input layer, we observe that selecting the first layer yields the worst performance, while the middle and the last layer yield better results. This trend resonates with the linear probing results presented in Figure A2–A4, showing that the proposed bridge input position selection strategy is reasonable. For the bridge output layer, no particular bridge output layers consistently yields the best performance, it varies for different bridge input layer. The selection of bridge input layer appears to have more impact on the downstream performance, which is reasonable, as the bridge input layer determines the quality and semantic richness of the intermediate representation that the bridge can leverage. In contrast, the bridge output layer primarily affects how the transformed features are reintegrated, leading to more nuanced variations.

### D.6 BRIDGE RANK, PROTOTYPE SET, DATASET SIZE ABLATION RESULTS

We conduct additional ablation studies on ISRUC (ECG → EEG) and FOG (EEG → EMG). Figures A9– A10 show that the bridge is relatively robust to the choice of bridge rank $r$, the number of prototypes $N_p$, and reduce the size of the training dataset $|\mathcal{D}^{(pair)}|$ %, maintaining stable performance in different parameter settings. Interestingly, we observe a sharp drop in performance when the size of the training dataset is reduced to 20% on FOG, but a relatively small performance drop on WESAD and ISRUC.

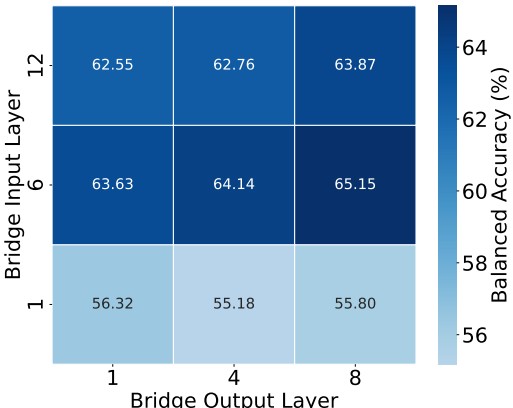

Figure A6: Bridge at nine predefined positions for ISRUC ECG → EEG. The results are averaged over five seeds. For reference, the proposed bridge position selection strategy selected input layer 12 and output layer 1 for 62.55% balanced accuracy.

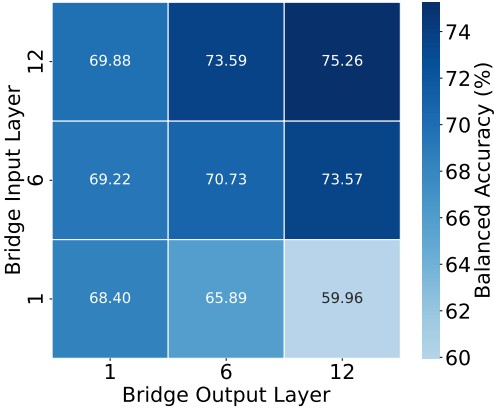

Figure A7: Bridge at nine predefined positions for FOG EEG → EMG. The results are averaged over five seeds. For reference, the proposed bridge position selection strategy selected input layer 4 and output layer 12 for 72.24% balanced accuracy.

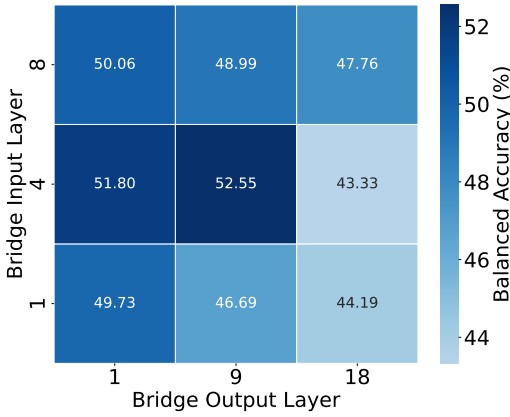

Figure A8: Bridge at nine predefined positions for WESAD (PPG → ECG). The results are averaged over five seeds. For reference, the proposed bridge position selection strategy selected input layer 7 and output layer 8 for 52.02% balanced accuracy.

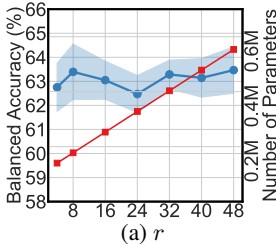 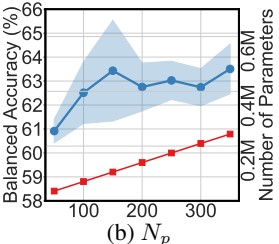 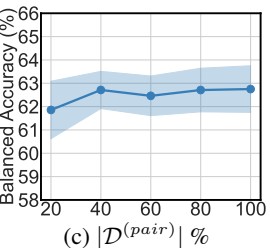

Figure A9: Bridge Training Ablation ISRUC (ECG → EEG). Blue: Balanced Accuracy. Red: Number of Parameters. We vary (a) bridge rank, (b) number of prototypes, and (c) pair dataset size to understand the robustness of BioX-Bridge and its performance under a low-data regime.

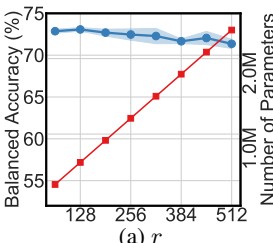 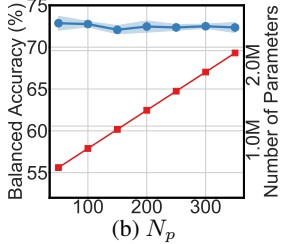 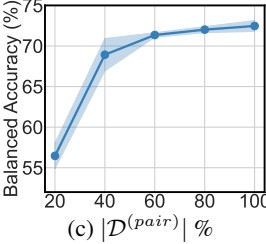

Figure A10: Bridge Training Ablation FOG (EEG → EMG). Blue: Balanced Accuracy. Red: Number of Parameters. We vary (a) bridge rank, (b) number of prototypes, and (c) pair dataset size to understand the robustness of BioX-Bridge and its performance under a low-data regime.

## D.7 BRIDGE ARCHITECTURE ABLATION

We conducted an additional experiment replacing the proposed bridge network (BioX-Bridge) with a fully connected layer (FC-Bridge) for WESAD (PPG → ECG). From Table A8, it is observed that FC-Bridge achieves worse transfer performance (47.21%) and requires more parameters (15.4M) compared to BioX-Bridge (52.62% and 0.4M). This highlights the necessity of a structured, low-rank bridge architecture rather than a naive fully connected alternative.

Table A8: Bridge architecture ablation. WESAD (PPG → ECG). We replace the proposed bridge network (BioX-Bridge) with a fully connected layer (FC-Bridge).

| Methods | Input Modality | Balanced Accuracy ↑ | F1 Macro ↑ | F1 Weighted ↑ | Trainable Parameters ↓ |
|---|---|---|---|---|---|
| Random | - | 33.33 | 31.29 | 35.38 | - |
| KD | ECG | 47.03±1.60 | 46.36±1.98 | 60.29±2.51 | 30.4M |
| KD-Contrast | ECG | 50.85±3.61 | 49.31±3.13 | 63.72±3.22 | 30.4M |
| BioX-Bridge | ECG | **52.02±0.80** | **52.62±0.36** | **65.12±0.91** | **0.4M** |
| FC-Bridge | ECG | 48.59±1.18 | 47.21±1.59 | 61.75±1.90 | 15.4M |
| Oracle | PPG | 62.96 | 60.97 | 74.52 | - |

## D.8 BRIDGE WITH RESHUFFLED SUBJECTS

To examine how BioX-Bridge performance varies when considering different subjects, we reshuffle the subjects and repeat the dataset splitting procedure. Then, we re-run the experiments for WESAD PPG → ECG using the same hyperparameters as before.

Table A9 presents the results for the reshuffled split. We see that BioX-Bridge continues to outperform the baselines while significantly reducing the trainable parameters. Interestingly, we see that for the reshuffled split, KD-Contrast performs worse than KD, which is not the case for the original split.

When comparing the two splits (original split Table A3b vs reshuffled split Table A9), we observe that BioX-Bridge achieves similar balanced accuracy (52.02% vs 54.14%) and F1-macro (52.62% vs 49.17%). However, there is a significant drop in F1-weighted (65.12% vs 52.18%). A similar drop is observed for the knowledge distillation baselines. The drop in F1-weighted can be explained by the fact that the Oracle (PPG) or the teacher for knowledge distillation, also observed a significant drop in F1-weighted (74.52% vs 53.71%).

Overall, we see that there are performance differences when considering different subjects, and it is largely a result of differences in performance of the Oracle models. This is a consequence inherent to the knowledge transfer problem itself.

Table A9: Bridge with reshuffled subject. WESAD (PPG → ECG). We reshuffle the subjects and repeat the dataset splitting procedure.

| Methods | Input Modality | Balanced Accuracy ↑ | F1 Macro ↑ | F1 Weighted ↑ | Trainable Parameters ↓ |
|---|---|---|---|---|---|
| Random | - | 33.33 | 31.33 | 35.33 | - |
| KD | ECG | 48.31±4.10 | 42.53±5.41 | 49.67±2.53 | 30.4M |
| KD-Contrast | ECG | 45.05±0.99 | 41.52±2.38 | 42.20±5.06 | 30.4M |
| BioX-Bridge | ECG | **54.13±1.15** | **49.17±0.84** | **52.18±0.75** | **0.4M** |
| Oracle | PPG | 65.92 | 54.74 | 53.71 | - |
| Oracle | ECG | 55.93 | 51.74 | 62.92 | - |

## APPENDIX REFERENCES

Yamini Bansal, Preetum Nakkiran, and Boaz Barak. Revisiting model stitching to compare neural representations. *NeurIPS*, 34:225–236, 2021.

Adrián Csiszárik, Péter Kőrösi-Szabó, Akos Matszangosz, Gergely Papp, and Dániel Varga. Similarity and matching of neural network representations. *NeurIPS*, 34:5656–5668, 2021.

Ching Fang, Christopher Sandino, Behrooz Mahasseni, Juri Minxha, Hadi Pouransari, Erdrin Azemi, Ali Moin, and Ellen Zippi. Promoting cross-modal representations to improve multimodal foundation models for physiological signals. *arXiv preprint arXiv:2410.16424*, 2024.

Haoyu He, Zizheng Pan, Jing Liu, Jianfei Cai, and Bohan Zhuang. Efficient stitchable task adaptation. In *CVPR*, pp. 28555–28565, 2024.

Karel Lenc and Andrea Vedaldi. Understanding image representations by measuring their equivariance and equivalence. In *CVPR*, pp. 991–999, 2015.

Feng Liu, Guangquan Zhang, and Jie Lu. Heterogeneous domain adaptation: An unsupervised approach. *IEEE transactions on neural networks and learning systems*, 31(12):5588–5602, 2020.

Ran Liu, Ellen L Zippi, Hadi Pouransari, Chris Sandino, Jingping Nie, Hanlin Goh, Erdrin Azemi, and Ali Moin. Frequency-aware masked autoencoders for multimodal pretraining on biosignals. *arXiv preprint arXiv:2309.05927*, 2023.

Luca Moschella, Valentino Maiorca, Marco Fumero, Antonio Norelli, Francesco Locatello, Emanuele Rodola, et al. Relative representations enable zero-shot latent space communication. In *ICLR*, 2023.

Zizheng Pan, Jianfei Cai, and Bohan Zhuang. Stitchable neural networks. In *CVPR*, pp. 16102–16112, 2023.

Maurice Rohr, Christoph Reich, Andreas Höhl, Timm Lilienthal, Tizian Dege, Filip Plesinger, Veronika Bulkova, Gari Clifford, Matthew Reyna, and Christoph Hoog Antink. Exploring novel algorithms for atrial fibrillation detection by driving graduate level education in medical machine learning. *Physiological Measurement*, 43(7):074001, 2022.

Garrett Wilson and Diane J Cook. A survey of unsupervised deep domain adaptation. *ACM Transactions on Intelligent Systems and Technology (TIST)*, 11(5):1–46, 2020.

Jiawen Yang, Shuhao Chen, Yucong Duan, Ke Tang, and Yu Zhang. Heterogeneous-modal unsupervised domain adaptation via latent space bridging. *arXiv preprint arXiv:2506.15971*, 2025.

Xingyi Yang, Daquan Zhou, Songhua Liu, Jingwen Ye, and Xinchao Wang. Deep model reassembly. *NeurIPS*, 35:25739–25753, 2022.

Wei Zhang, Zhuokun Yang, Hantao Li, Debin Huang, Lipeng Wang, Yanzhao Wei, Lei Zhang, Lin Ma, Huanhuan Feng, Jing Pan, et al. Multimodal data for the detection of freezing of gait in parkinson's disease. *Scientific data*, 9(1):606, 2022a.

Xiang Zhang, Ziyuan Zhao, Theodoros Tsiligkaridis, and Marinka Zitnik. Self-supervised contrastive pre-training for time series via time-frequency consistency. *Advances in neural information processing systems*, 35:3988–4003, 2022b.

