# OpenReview forum: "BioX-Bridge: Model Bridging for Unsupervised Cross-Modal Knowledge Transfer across Biosignals"
_ICLR.cc/2026/Conference — ICLR 2026 Oral_

### Official Review · Reviewer_mvhb · 2025-10-26

**Soundness:** 4
**Presentation:** 4
**Contribution:** 3
**Rating:** 8
**Confidence:** 4

**Summary:**

This paper researches in the domain of multimodal biosignals, investigating how to transform the knowledge across different biosignal foundation models/modalities while being parameter efficient. Towards this end, the paper proposes to train a lightweight bridge network to align representations across modalities. Specifically, the authors investigated (1) How to select which intermediate representations to use for the construction of the bridge network; (2) Using prototype network with low-rank approximation to guide the learning of the bridge network. The proposed architecture, BioX-Bridge, is validated on three multimodal biosignal datasets, showing better or comparable transfer performance while significantly reducing the parameters needed.

**Strengths:**

1. The paper investigated an important problem of multimodal biosignal knowledge transfer.
2. Novel idea, the bridge architecture is nicely designed with the use of (1) pseudo labels; (2) linear CKA. The bridge architecture also uses a prototype network, that increases the flexibility of incorporating previous knowledges.
3. Detailed experimental design. Solid selection of existing backbone models for each modality and baseline architectures. The model performs decently well considering its parameter size.
4. Insightful ablation experiments. Especially insightful to see how the total amount of prototypes would affect the performance, and how the designed architecture is robust to total number of samples.

**Weaknesses:**

1. The experimental design can be more comprehensive. The authors applied the proposed architecture on a total of three datasets, in each one, they consider 2 transfer directions. The paper lacks an analysis on how the model would perform in situations where the dataset contains more than 2 modalities, for example, check the datasets [1] and experiments presented in [2, 3]. Also, it is unclear how the proposed bridge architecture is specific to each dataset - an interesting experiment to consider is to consider two datasets that share the same multimodality e.g. ECG and EEG, and investigate how the bridge encoder differ for two datasets.

[1] Zhang, Xiang, Ziyuan Zhao, Theodoros Tsiligkaridis, and Marinka Zitnik. "Self-supervised contrastive pre-training for time series via time-frequency consistency." Advances in neural information processing systems 35 (2022): 3988-4003.

[2] Liu, Ran, Ellen L. Zippi, Hadi Pouransari, Chris Sandino, Jingping Nie, Hanlin Goh, Erdrin Azemi, and Ali Moin. "Frequency-aware masked autoencoders for multimodal pretraining on biosignals." arXiv preprint arXiv:2309.05927 (2023).

[3] Fang, Ching, Christopher Sandino, Behrooz Mahasseni, Juri Minxha, Hadi Pouransari, Erdrin Azemi, Ali Moin, and Ellen Zippi. "Promoting cross-modal representations to improve multimodal foundation models for physiological signals." arXiv preprint arXiv:2410.16424 (2024).

2. Completeness of existing experimental results. The paper lacks a more detailed explanation of how the prototypes are selected. Also, for the ablation experiments, the ablation is only applied on the WESAD dataset. Are the conclusions consistent across different datasets?

**Questions:**

1. I am particularly curious about the distribution of the elements computed before the argmin inside equation [4] and [5]. Specifically, how different are the input intermediate representations across different layers for each dataset? How different are the similarity of representation in each layer $l \in \[1, ..., L\]$ for a given layer m? Also, how would the value of m and l differ when considering the same dataset but different sets of samples (e.g. would it be better if the selection of m and l are data-dependent)?

2. Does the performance differ a lot when considering different subjects?

---

> ### Author Response · Authors · 2025-11-21
> **Response to Reviewer mvhb (Part 1)**
>
> We sincerely appreciate the reviewer’s thoughtful comments and the recognition of the strengths in our work. Below, we address each question and weakness in detail. All corresponding revisions have been highlighted in red in the updated manuscript.
>
> ### Weakness 1. Additional modalities and datasets
>
> We agree that the suggested analyses would provide valuable and interesting insights as extensions to our evaluation. However, due to computational limitations and the short turnaround time for the rebuttal, we were unable to complete them by the recommended deadline. We prioritized the remaining experiments suggested by the reviewer that are most central to our core contributions, including the completeness of existing experimental results, the ablation study on bridge position selection, and the robustness analyses across subjects. Nevertheless, we acknowledge the merits of these ideas and have added a detailed discussion in Appendix B outlining how we plan to extend our method and investigate these questions in light of the suggested papers.
>
> ### Weakness 2. Prototype selection and ablation for FOG+ISRUC
>
> To initialize the learnable prototype set, $ \mathbf{P} \in \mathbb{R}^{N_p \times d^{(\mathrm{old})}_l} $, we randomly select $N_p$ token/feature maps from $\mathbf{h}^{(\mathrm{old})}_l $ of the training set samples. Specifically, $\mathbf{h}^{(\mathrm{old})}_l $ are the intermediate representations from the $l$-th layer of the old modality model, which are also the alignment targets for the bridge output. We have updated Section 3.4 to clarify this procedure.
>
> In addition to WESAD, we conducted additional ablation studies on ISRUC and FOG in Appendix D.6 Bridge Rank, Prototype Set, Dataset Size Ablation Results. Figure A9-A10 shows that BioX-Bridge is relatively robust to the choice of bridge rank $r$, the number of prototypes $N_p$, and reduced training dataset size $|\mathcal{D}^{(pair)}|$ \%, maintaining stable performance across different parameter settings. The results are consistent with WESAD, except for one case: when the training dataset size is reduced to 20%, a sharp drop in performance is observed on FOG, but a relatively small performance drop is observed on WESAD and ISRUC.

---

> > ### Author Response · Authors · 2025-11-21
> > **Response to Reviewer mvhb (Part 2)**
> >
> > ### Question 1. Bridge position selection
> >
> > We present the layer-wise results for the bridge position selection on all three datasets and six directions in Appendix D.4.
> >
> > **Bridge input position selection:** From Figure A2-A4, it is observed that the pseudo label linear probing performance using representations from different layer $m$ of $f_\phi^{\text{(new)}}$ varies quite significantly. The linear probing performance range (max-min) across different layer $m$ is as low as 2\% for FOG EEG $\rightarrow$ EMG and as high as 14\% for WESAD ECG $\rightarrow$ PPG. Therefore, the difference in input intermediate representations across different layers varies across datasets and transfer directions. However, this difference is not solely a result of dataset characteristics, but also influenced by the model architecture. In general, linear probing performance tends to increase as we retrieve representations from deeper layers of transformers (LaBraM, NormWear, HuBERT-ECG), and the same does not apply to CNN (PaPaGei), where we observed better linear probing performance in the shallower layers.
> >
> > **Bridge output position selection:** For a given layer $m$ from $f_\phi^{\text{(new)}}$, as we vary layer $l$ from $f_\theta^{\text{(old)}}$, we do not observe a clear trend for the similarity of representations between the two layers across the datasets and transfer directions studied. Note that the absolute CKA similarity, on the scale of 1e-2, is low across all datasets, where CKA ranges between 0 and 1. This is expected as the two models are trained on data from different modalities, and it is the relative difference in similarity between layer $l$ and a given layer $m$ that is of interest for bridge output position selection. Define the coefficient of variation (CV) for each transfer direction as:
> > \begin{equation}
> > \mathrm{CV} = \frac{\sigma}{\mu} \times 100
> >             = \frac{\sqrt{\frac{1}{L}\sum_{l=1}^{L} (c_l - \mu)^2}}{\frac{1}{L}\sum_{l=1}^{L} c_l} \times 100.
> > \end{equation}
> > where $c_l$ is the CKA similarity between representations from layer $m$ of $f_\phi^{\text{(new)}}$ and layer $l$ of $f_\theta^{\text{(old)}}$, and $\mu$ and $\sigma$ denote the mean and standard deviation of $\{c_l\}_{l=1}^{L}$, respectively. The CV is as low as 2.05 for WESAD ECG $\rightarrow$ PPG and as high as 30.72 for FOG EEG $\rightarrow$ EMG.
> >
> > **Different dataset subset for position selection:** We further analyse the impact of reduced dataset size on the bridge position selection results. Figure A5 shows bridge position selection results on WESAD when using 50% of the data compared to using 100% data in Figure A4. Comparing these two figures, we observe very similar trends for the linear probing and CKA similarity plots. For PPG $\rightarrow$ ECG, the same bridge input and output layers are selected. For ECG $\rightarrow$ PPG, the bridge input layer differs by one layer while the bridge output layer is the same. This suggests that the bridge position selection strategy is relatively stable with respect to the number of samples used in the dataset.
> >
> > While our current bridge position selection strategy already performs well in selecting an appropriate layer with minimal computational overhead, we agree that a data-free bridge position selection strategy would certainly help to further improve the flexibility of BioX-Bridge for different application scenarios. In light of this, we have added this discussion to Appendix B Limitations and Future Work.

---

> > > ### Author Response · Authors · 2025-11-21
> > > **Response to Reviewer mvhb (Part 3)**
> > >
> > > ### Question 2: Different subjects
> > >
> > > In our experiments, we split the dataset in a subject-independent manner, i.e. no data from any given subject appears in more than one split. To examine how performance varies when considering different subjects, we reshuffle the subjects and repeat the splitting procedure. Then, we re-run the experiments for WESAD PPG → ECG using the same hyperparameters as before. The following tables present the results for the original split and the reshuffled split.
> > >
> > > Looking at the reshuffled split, we see that BioX-Bridge continues to outperform the baselines while significantly reducing the trainable parameters. Interestingly, we see that for the reshuffled split, KD-Contrast performs worse than KD, which is not the case for the original split.
> > >
> > > When comparing the two splits (original split vs reshuffled split), we observe that BioX-Bridge achieves similar balanced accuracy (52.02% vs 54.14%) and F1-macro (52.62% vs 49.17%). However, there is a significant drop in F1-weighted (65.12% vs 52.18%). A similar drop is observed for the knowledge distillation baselines. The drop in F1-weighted can be explained by the fact that the Oracle (PPG) or the teacher for knowledge distillation, also observed a significant drop in F1-weighted (74.52% vs 53.71%).
> > >
> > > Overall, we see that there are performance differences when considering different subjects, and it is largely a result of differences in the performance of the Oracle models. This is a consequence inherent to the knowledge transfer problem itself.
> > >
> > > **Table. Original Split**
> > >
> > > | Dataset: WESAD  | PPG (Old) -> ECG (New) |  |  |  |
> > > |---|:---:|:---:|:---:|:---:|
> > > | Methods (Modality Used) | Balanced Accuracy | F1-Macro | F1-Weighted | Parameters |
> > > | Random | 33.33 | 31.29 | 35.38 | - |
> > > | KD (ECG) | 47.03 | 46.36 | 60.29 | 30.4M |
> > > | KD-Contrast (ECG) | 50.85 | 49.31 | 63.72 | 30.4M |
> > > | BioX-Bridge (ECG) | 52.02 | 52.62 | 65.12 | 0.4M |
> > > | Oracle (PPG) | 62.96 | 60.97 | 74.52 | - |
> > > | Oracle (ECG) | 49.47 | 51.05 | 62.48 | - |
> > >
> > > **Table. Reshuffled Split**
> > >
> > > | Dataset: WESAD  | PPG (Old) -> ECG (New) |  |  |  |
> > > |---|:---:|:---:|:---:|:---:|
> > > | Methods (Modality Used) | Balanced Accuracy | F1-Macro | F1-Weighted | Parameters |
> > > | Random | 33.33 | 31.33 | 35.33 | - |
> > > | KD (ECG) | 48.31 | 42.53 | 49.67 | 30.4M |
> > > | KD-Contrast (ECG) | 45.05 | 41.52 | 42.20 | 30.4M |
> > > | BioX-Bridge (ECG) | 54.13 | 49.17 | 52.18 | 0.4M |
> > > | Oracle (PPG) | 65.92 | 54.74 | 53.71 | - |
> > > | Oracle (ECG) | 55.93 | 51.74 | 62.92 | - |
> > >
> > > We have added the above results and discussions to Appendix D.8 Bridge with Reshuffled Subjects.

---

> > > > ### Author Response · Authors · 2025-12-03
> > > > **Response to Reviewer mvhb (Follow-up)**
> > > >
> > > > Following the suggestion on how the model would perform when the dataset contains a third modality, we had the opportunity to conduct experiments using an additional modality, skin conductance (SC), for the FOG dataset. The results are summarized in the table below. We see that BioX-Bridge continues to match/outperform baselines while significantly reducing the number of trainable parameters.
> > > >
> > > > | Dataset: FOG  | EEG (Old) -> SC (New) |  |  |  |
> > > > |---|:---:|:---:|:---:|:---:|
> > > > | Methods (Modality Used) | Balanced Accuracy | F1-Macro | F1-Weighted | Parameters |
> > > > | Random | 50.00 | 49.99 | 50.01 | - |
> > > > | KD (SC) | 56.15 | 53.43 | 53.48 | 136.1M |
> > > > | KD-Contrast (SC) | 58.09 | 58.06 | 58.07 | 136.1M |
> > > > | BioX-Bridge (SC) | 59.30 | 58.95 | 59.01 | 1.2M |
> > > > | Oracle (EEG) | 72.15 | 72.14 | 72.20 | - |
> > > > | Oracle (SC) | 65.69 | 65.57 | 65.67 | - |

---

### Official Review · Reviewer_iQEx · 2025-10-31

**Soundness:** 3
**Presentation:** 3
**Contribution:** 2
**Rating:** 4
**Confidence:** 3

**Summary:**

The paper presents a parameter-efficient framework for unsupervised cross-modal knowledge transfer across heterogeneous modalities. The proposed BioX-Bridge trains a lightweight “bridge” network to align intermediate representations between pre-trained foundation models from different biosignal modalities (e.g., ECG, EEG, PPG, EMG), enabling knowledge transfer without requiring labeled data from the target modality. The framework comprises two key components: a two-stage strategy for selecting the optimal bridge positions (in the source and target network) and a prototype-based architecture with low-rank approximation for efficient high-dimensional projection. The authors evaluate BioX-Bridge across three datasets (WESAD, FOG, ISRUC) with four modalities and six transfer directions, showing that it achieves performance comparable to/better than knowledge distillation baselines while significantly reducing trainable parameters.

**Strengths:**

- Leveraging large foundation models for efficient cross-modal knowledge transfer without the computational overhead of “standard” knowledge distillation.
- The design is straightforward and allows flexibility, incorporating different foundation models as backbones.
- The paper is well-organized, well-motivated, and easy to follow.

**Weaknesses:**

- W1: Performance improvements. I agree with the authors' motivation that there are domains/modalities where data is not readily available for (pre-)training large models, in which case approaches like BioX-Bridge can be useful. However, in this particular case, it seems that performance degrades when using BioX-Bridge compared to both source/target oracles. Looking at Table 1, in all but one case, applying BXB leads to worse performance than training a model directly on the “target” modality. This is not explicitly discussed in the paper, so I am comparing across the table (eg. ECG (New) 60.11 vs ECG (Oracle) 63.54). This opens a practical question if, at least for these applications, there is a need for such bridging, and if there is, will it lead to benefits.

- W2. The authors state (L355) " our method for efficient cross-modal knowledge transfer would be even more valuable as they scale up”. While this may be true in terms of the bridge architecture, in general, I find this counterintuitive in how the bridge in/out placements are chosen. As it is, stage 1, involves exhaustive search of the appropriate layer: This would become really expensive with larger models and more sophisticated probes. Therefore, BXB might not scale well overall.

**Questions:**

- See weakness

- If the goal is to align representations between layer m (target modality) and layer l (source modality), why propagate both to the final layer for alignment (Eq. 7)? It seems more direct to use $h^{(old)}_l$ as the pseudo label and align to the prediction/output of $h^{(new)}_m$  This design choice needs clarification—does propagating to the final layer improve alignment?

- The ablation study wrt the bridge placement needs more discussion and detail. On average, choosing a fixed position doesn’t seem to lead to a substantial performance degradation. Are the performances of the “fixed” positions consistent across datasets/modalities/models? Are some combinations better than others?

- How does the bridge position selection strategy perform when the foundation models have very different architectures? The current evaluation uses mostly transformer variants, but biosignal models span can diverse architectures (and modalities).

- Minor comment: I found the notations of “old” and “new” a tad confusing. I would suggest the authors consider using “source/target” for the different models (this is also how I refer to it in the comments)

---

> ### Author Response · Authors · 2025-11-21
> **Response to Reviewer iQEx (Part 1)**
>
> We sincerely thank the reviewer for their thoughtful and detailed evaluation of our work. Below, we address each question and weakness in detail. All corresponding revisions have been highlighted in red in the updated manuscript.
>
> ### Weakness 1: Performance improvements
>
> We would like to clarify an important distinction between the oracle models and BioX-Bridge. The oracle results correspond to **supervised training** using labeled data from the new modality, whereas BioX-Bridge operates in a **fully unsupervised** setting, where no labeled data from the new modality are available. Thus, the oracle and BioX-Bridge are designed for fundamentally different application scenarios. The oracle represents an upper bound achievable when labeled data exist, while BioX-Bridge addresses the realistic and often more challenging case where such labels are absent.
>
> Consequently, BioX-Bridge is not expected to outperform supervised oracles but instead aims to match the oracle results. The modest drop in performance compared to supervised oracles is therefore expected and acceptable. Moreover, BioX-Bridge consistently outperforms/matches other baselines (e.g., KD, KD-Contrast) with a significantly reduced number of parameters, demonstrating its effectiveness within the intended setting.
>
> We agree that this distinction should be made more explicit in the paper. We highlight this in Section 4.2 Unsupervised Cross-Modal Knowledge Transfer Performance, and have updated it in light of the discussion.
>
> ### Weakness 2. Efficiency of BioX-Bridge as foundation models scale up
>
> As the foundation models scale up, BioX-Bridge is efficient in terms of the bridge architecture to help reduce the number of trainable parameters, enabling efficient cross-modal knowledge transfer without access to high-end GPUs. The reviewer also correctly points out that as the foundation models scale up, there would be more layers to choose from, which would require more computation to select the bridge in/out placements.
>
> Fortunately, the bridge position selection strategy is lightweight in comparison to knowledge distillation and scales linearly. Specifically, we perform a single forward inference with $f_\theta^{\text{(old)}}$ and $f_\phi^{\text{(new)}}$ to cache the intermediate representations and pseudo labels. Subsequently, linear probing identifies the bridge input position, and CKA similarity is computed to select the bridge output position. Therefore, the bridge position selection strategy scales linearly with the number of layers from the foundation models, as opposed to quadratically if we were to study every possible combination of bridge input and output positions. Furthermore, it only takes a few minutes to cache the representation and less than half an hour to select the bridge input and output positions on CPU, in comparison to hours of training for knowledge distillation or bridge on GPU. Thus, the selection procedure remains computationally efficient and practical even for large-scale models, with plenty of room for acceleration if needed.
>
> We have updated Section 4.1 Backbone Foundation Models to reflect the discussion above.

---

> > ### Author Response · Authors · 2025-11-21
> > **Response to Reviewer iQEx (Part 2)**
> >
> > ### Question 1. Choice of layer for loss calculation
> >
> > In our preliminary experiments, we performed the alignment at the $l$-th layer, like the reviewer suggested. However, we found that the downstream performance remained poor, where the model predictions collapsed to one class, even though the alignment loss is low. Interestingly, when we propagate to the last layer and perform alignment at the $L$-th (final) layer, the downstream performance is much better. We believe this is a result of error propagation. For alignment at the $l$-th layer, a small alignment error would grow as it propagates to the final layer and the classifier. For alignment at the $L$-th layer, the error growth will be reflected in the alignment loss, enabling the bridge to take that into account and yield better downstream performance.
> >
> > We have updated the footnote on Line 377 to reflect this discussion.
> >
> > ### Question 2. Bridge placement ablation
> >
> > To better understand the impact of bridge placement, we provide a breakdown of the performance at each of the nine combinations from the bridge placement ablation table (Table 2) in the form of a heatmap in Figure A8 for WESAD. To understand variations across datasets, we repeat the same experiment for ISRUC (Figure A6) and FOG (Figure A7).
> >
> > From these figures, we observe that there does not exist a particular position that consistently outperforms other positions **across datasets/modalities/models**. In terms of the bridge input layer (new model layer), we observe that selecting the first layer yields the worst performance, while the middle and the last layer yield better results. For the bridge output layer (old model layer), no particular bridge output layer consistently yields the best performance; it varies for different bridge input layers. The selection of the bridge input layer appears to have more impact on the downstream performance, which is reasonable, as the bridge input layer determines the quality and semantic richness of the intermediate representation that the bridge can leverage. In contrast, the bridge output layer primarily affects how the transformed features are reintegrated and aligned with the old modality model, leading to more nuanced variations.
> >
> > We have added a discussion to Appendix D.5 Bridge Position Ablation Results to reflect the observations above.
> >
> > ### Question 3: Backbone architecture and bridge position selection strategy
> >
> > We would like to clarify that the current evaluations presented in Table 1 cover diverse biosignal foundation model architectures, including CNN-transformer (LaBraM, HuBERT-ECG, NormWear) and CNN (PaPaGei). To the best of our knowledge, existing open-source biosignal foundation models adopt one of the aforementioned architecture variants. To further showcase that BioX-Bridge generalizes to traditional biosignal models, we showcase additional results replacing the HuBERT-ECG foundation model with ECG-DualNet, a CNN-LSTM model, in Appendix Table A6.
> >
> > In terms of the bridge position selected for different foundation model architectures, we added the bridge input and output position selection results to Appendix D.4.
> >
> > For the bridge input position, linear probing with pseudo labels appears to perform best in the latter layers for transformer-based models, and it is not the case for CNN-based PaPaGei. A possible explanation for this difference lies in how the two architectures organize information across different layers. For CNN-based models, earlier layers capture fine-grained and low-level features, while the later layers capture coarse and high-level features with larger receptive fields. The lower-level features might be more informative for the task. Unlike CNN, transformer-based models have access to the same receptive field across all layers. As the model continues to refine and reorganize the representations across layers, the latter representation yields better linear probing performance.
> >
> > For the bridge output position, the CKA similarity difference across different layers of the model appears to vary based on the dataset. Overall, the selection of the bridge input layer appears to have more impact on the downstream performance, which is reasonable, as the bridge input layer determines the quality and semantic richness of the intermediate representation that the bridge can leverage. In contrast, the bridge output layer primarily affects how the transformed features are reintegrated and aligned with the old modality model, leading to more nuanced variations.

---

> > > ### Author Response · Authors · 2025-11-21
> > > **Response to Reviewer iQEx (Part 3)**
> > >
> > > ### Question 4: Notation for old/new vs. source/target
> > >
> > > In fact, in the initial draft of our paper, we opted to use the source/target notation to align with transfer learning and domain adaptation terminologies. However, during the revision of the paper, we found that it may be a bit misleading in our specific setting due to the presence of two opposite directions of flow. On one hand, the knowledge transfer proceeds from source to target, where the goal of the framework is to transfer knowledge from a well-trained model for the source modality to another model for the target modality. On the other hand, the bridge processes information from target to source, where the bridge takes representations from the target modality as input and projects them into the representation space of the source modality as output. If we opt for source/target notation, Equation 2 becomes
> > > $$
> > > \boldsymbol{\tilde{h}}^{\text{(source)}}\_{l} = b_{\boldsymbol{\psi}} \left( \boldsymbol{h}^{\text{(target)}}\_{m} \right)
> > > $$
> > >
> > > The bridge receives input from the target modality and outputs to the source modality, which is somewhat awkward and counterintuitive, as one would normally expect a model to process the source as input and output the target. Therefore, we opted to adopt the old/new notation, in hopes of avoiding the bias behind the meaning of target/source. Furthermore, old/new is more intuitive in relation to the problem setup, where we have models from an existing old modality and would like to develop a new model for the new modality. We hope this clarifies our motivation behind using the old/new notation and would love to hear your thoughts and make further changes.

---

> > > > ### Comment · Reviewer_iQEx · 2025-11-26
> > > >
> > > > I thank the authors for their detailed and thoughtful response and additional analysis included in the revised manuscript. I appreciate the effort to address my concerns. Overall, the authors addressed most of my concerns, and as such, I will increase my score.

---

### Official Review · Reviewer_Yb6i · 2025-10-31

**Soundness:** 3
**Presentation:** 3
**Contribution:** 2
**Rating:** 6
**Confidence:** 3

**Summary:**

The authors are focused on the problem of cross-modal transfer between biosignals, broadly trying to distill knowledge from one modality into another. They introduce their model BioX-Bridge, which attempts to learn a transformation from intermediate representations between two frozen foundation models. They argue that this is a more parameter efficient way of doing cross-modal transfer than previous methods. The model works as following: (1) use linear probing to find the layer in your new modality model that maintains good probing performance on labels from the old modality model. (2) select the layer in the old modality model that you want to conduct the transfer from by maximizing CKA similarity to the layer identified in the previous step. (3) train a two-module network that predicts representations from the selected old modality layer given representations from the selected new modality layer. They conduct experiments across three different datasets and compare to other baselines. They show that BioX-Bridge does comparably or better than baselines, and make the point that training parameters are much less than the baselines.

**Strengths:**

- Authors do a good job of discussing prior paper and contextualizing their paper in the broader literature. Figures are also clear and informative.
- The problem is well-motivated and has clear relevance to practical applications, in particular when discussing parameter efficiency.
- Experiments show a nice diversity of modality choices (EEG/ECG, EEG/EMG, ECG/PPG)
- Authors show a good number of different ablations and baseline comparisons.

**Weaknesses:**

- A key part of the method is the choice of bridge layers. To choose the bridge layer in the new modality, the authors say they find the layer that best linearly represents pseudo labels from the old modality. My understanding is that the pseudo labels used are in fact the downstream task labels. I wonder if this gives their method an advantage over other baseline models that should be discussed. That is, the parameter efficiency of BioX-Bridge is because mappings are learned only between two carefully chosen layers. However, if this choice process relies on the use of downstream task labels, this means the bridge positions are not task-agnostic and that the efficiency comes from having access to the downstream labels. Are the other baseline methods (KD, KD-Contrast, etc) more task agnostic? It seems like these points should be discussed and clarified.

**Questions:**

- I'm confused by some details of the bridge architecture. It seems like the bridge is learned per-token-- e.g., as mentioned in section 3.4 if the bridge were a linear map there would be $N_{m}^{new} \times d_{m}^{new} \times N_{l}^{old} \times d_{l}^{old}$ parameters, not $d_{m}^{new} \times d_{l}^{old}$ parameters. Why this choice of mapping? This seems like an unnecessary constraint too, since the input time series may not be naturally chunked into sequence inputs. Did you all try learning a mapping that isn't token specific, e.g. a simple MLP?
- I would like a better understanding of how much the bridge selection process affects the performance. Table 2 kind of gets at this, but since it's averaged over 9 different combinations it's hard to interpret. Maybe just separate it out, and also show the linear probe performance for the new model layer for each of them.

---

> ### Author Response · Authors · 2025-11-21
> **Response to Reviewer Yb6i (Part 1)**
>
> We sincerely thank the reviewer for engaging deeply with our work and for identifying important directions to clarify and expand the evaluation and discussion. Below, we address each question and weakness in detail. All corresponding revisions have been highlighted in red in the updated manuscript.
>
> ### Weakness 1. Use of pseudo label and task-agnosticism
>
> The bridge position selection process consists of two stages: bridge input position selection and bridge output position selection. The bridge input position selection (bridge layer in the new modality) can be formulated as:
> $$
> \arg \text{min}\_{m \in \{1, \dots, M\}} \frac{1}{|\mathcal{D}^{\text{(pair)}}|} \sum_{i=1}^{|\mathcal{D}^{\text{(pair)}}|} \mathcal{L}\_{\text{probe}} \left(g_{\boldsymbol{\eta}}\left(\boldsymbol{h}\_{m,i}^{\text{(new)}}\right), \hat{y}\_i \right)
> $$
> where $\boldsymbol{h}\_{m,i}^{\text{(new)}} = f^{\text{(new)}}\_{\boldsymbol{\phi}\_{\leq m}}(\mathbf{x}\_i^{\text{(new)}})$ denotes the $i$-th sample's intermediate representation from the $m$-th layer of the new modality model, and $\hat{y}\_i = g\_{\boldsymbol{\omega}}^{\text{(old)}} \circ f^{\text{(old)}}\_{\boldsymbol{\theta}}(\mathbf{x}\_i^{\text{(old)}})$ denotes the pseudo label. $\mathcal{L}\_{\text{probe}}$ denotes the empirical loss for the linear prober $g\_{\boldsymbol{\eta}}$. **Importantly, the pseudo labels $\hat{y}\_i $ are derived from the old modality model $f^{\text{(old)}}\_{\boldsymbol{\theta}}$ and its classifier $g_{\boldsymbol{\omega}}^{\text{(old)}}$. Therefore, the pseudo labels are not the ground truth downstream task labels.** In fact, the pseudo labels we refer to here are simply the argmax of the teacher logits used in knowledge distillation. During knowledge distillation, the student model is updated such that the student logits mimic those of the teacher logits. **As a result, the use of pseudo labels does not give BioX-Bridge an advantage over other baseline methods, as the information is equally available to other baselines, ensuring fair comparisons.**
>
> We have modified Section 3.3 Bridge Position Selection to reflect the discussion above.
>
> For the current setup, since teacher logits/pseudo labels are used, BioX-Bridge and KD/KD-Contrast are not task-agnostic. CardioGAN, on the other hand, is task-agnostic, as it translates the data between modalities. Whether the cross-modal knowledge transfer method should be task-agnostic or not raises interesting discussion points. A task-agnostic method would offer better generality, making it suitable for multi-task scenarios. However, such approaches can potentially limit task-specific performance compared to task-specific methods. This is evident in the case of CardioGAN, where the ECG generated by CardioGAN from PPG has been shown to be capable of providing reliable heart rate measurements in the previous work [1], but its cross-modal knowledge transfer performance for WESAD is quite poor in our experiments. We believe this is an interesting next step to further improve the flexibility of BioX-Bridge for different application scenarios, where we investigate other bridge position selection strategies that are task-agnostic.
>
> We have modified Appendix B Limitations and Future Work to reflect the discussion above.
>
> References
>
> [1] Sarkar, Pritam, and Ali Etemad. "CardioGAN: Attentive generative adversarial network with dual discriminators for synthesis of ecg from ppg." Proceedings of the AAAI Conference on Artificial Intelligence. Vol. 35. No. 1. 2021.

---

> ### Author Response · Authors · 2025-11-21
> **Response to Reviewer Yb6i (Part 2)**
>
> ### Question 1. Bridge architecture and parameters
>
> The output shape of the bridge network is constrained by the need to maintain shape compatibility with the intermediate representations of the old modality biosignal model $f\_\theta^{\text{(old)}}$. Specifically, the bridge output must match the shape $N^{\text{(old)}}\_{l} \times d^{\text{(old)}}\_{l}$ for each sample, where $N^{\text{(old)}}\_{l}$ is the number of token (for transformers) or filters (for CNNs), and $d^{\text{(old)}}\_{l}$ is the embedding dimension. If the bridge output shape differs from this, a dimensional mismatch error would occur when connecting to the old modality model. While it is true that for variants of the transformer architecture, it would be sufficient to match only the embedding dimension $d^{\text{(old)}}\_{l}$, but for other architectures like CNN, a strict match is required. To ensure generalizability of the method across different model architectures, we do not place any simplification assumptions for the bridge output shape.
>
> As for the bridge input shape, we can assume that a reduction step can be performed (e.g. mean-pooling or use of a [CLS] token) to reduce the input shape from $N^{\text{(new)}}\_{m} \times d^{\text{(new)}}\_{m}$ to $d^{\text{(new)}}\_{m}$. In this case, the minimum number of parameters required for a linear projection would be $N^{\text{(old)}}\_{l} \times d^{\text{(old)}}\_{l} \times d^{\text{(new)}}\_{m} = 93 \times 512 \times 200 \approx 9.5$ million parameters for LaBram as $f\_\phi^{\text{(new)}}$ and HuBERT-ECG as $f\_\theta^{\text{(old)}}$, which exceeds the parameter count of LaBraM (5.8M).
>
> We conducted an additional experiment replacing the proposed bridge network (BioX-Bridge) with a fully connected layer (FC-Bridge) for WESAD PPG $\rightarrow$ ECG. It is observed that the FC-Bridge achieves worse transfer performance (47.21%) and requires more parameters (15.4M) compared to BioX-Bridge (52.62% and 0.4M).
>
> | Dataset: WESAD  | PPG (Old) -> ECG (New) |  |  |  |
> |---|:---:|:---:|:---:|:---:|
> | Methods (Modality Used) | Balanced Accuracy | F1-Macro | F1-Weighted | Parameters |
> | Random | 33.33 | 31.29 | 35.38 | - |
> | KD (ECG) | 47.03 | 46.36 | 60.29 | 30.4M |
> | KD-Contrast (ECG) | 50.85 | 49.31 | 63.72 | 30.4M |
> | BioX-Bridge (ECG) | 52.02 | 52.62 | 65.12 | 0.4M |
> | FC-Bridge (ECG) | 48.59 | 47.21 | 61.75 | 15.4M |
> | Oracle (PPG) | 62.96 | 60.97 | 74.52 | - |
> | Oracle (ECG) | 49.47 | 51.05 | 62.48 | - |
>
> We have added Appendix D.7 Bridge Architecture Ablation to include the experiment above.
>
> ### Question 2. Bridge position selection and downstream performance
>
> We provide a breakdown of the performance at each of the nine combinations from Table 2 in the form of a heatmap in Figure A8 for the WESAD dataset. We also plot the linear probing performance for each of the new model layer in Figure A4. To understand variations across datasets, we repeat the same experiments for ISRUC (Figure A6+A2) and FOG (Figure A7+A3).
>
> We can make a few observations based on the heatmaps and linear probe plots. In terms of the bridge input layer (new model layer), we observe that selecting the first layer yields the worst performance, while the middle and the last layer yield better results. This trend resonates with the linear probing results, showing that the proposed bridge input position selection strategy is reasonable. For the bridge output layer (old model layer), no particular bridge output layers consistently yield the best performance; it varies for different bridge input layers. The selection of the bridge input layer appears to have more impact on the downstream performance, which is reasonable, as the bridge input layer determines the quality and semantic richness of the intermediate representation that the bridge can leverage. In contrast, the bridge output layer primarily affects how the transformed features are reintegrated and aligned with the old modality model, leading to more nuanced variations.
>
> We have added discussion to Appendix D.4 Bridge Position Selection Results and Appendix D.5 Bridge Position Ablation Results to reflect the observations above.

---

### Author Response · Authors · 2025-12-03
**Summary of Rebuttal and Discussion**

Dear Reviewers and Area Chairs,

We express our deepest gratitude to all reviewers for their constructive feedback and engagement during the discussion phase.

We are encouraged by the recognition of the strengths of our work:
- Clear motivation and practical relevance of the problem (_Yb6i, iQEx, mvhb_)
- Novelty, efficiency, and flexibility of the proposed framework (_Yb6i, iQEx, mvhb_)
- Comprehensive experimental design and insightful ablations (_Yb6i, mvhb_)
- Clarity of presentation and organization (_Yb6i, iQEx, mvhb_)

We have modified the manuscript to address concerns from the reviewers:
- _Clarification on pseudo labels (Yb6i)_: We clarified that the pseudo labels used for bridge position selection are derived from teacher logits rather than ground truth labels, ensuring that BioX-Bridge operates under the same information constraints as other KD baselines.
- _Comparison between BioX-Bridge and oracle (iQEx)_: We clarified the relationship between oracle (supervised) and BioX-Bridge (unsupervised). We highlighted that BioX-Bridge achieves results comparable to supervised training, and that the results align with our understanding of how certain modalities are inherently better suited for particular tasks due to physiological relevance.
- _Scalability of BioX-Bridge (iQEx)_: We addressed the concern regarding the computational cost of bridge position selection as models scale up. We clarified that the selection strategy scales linearly with model depth, requiring negligible computational costs (minutes on CPU) compared to bridge training or knowledge distillation (hours on GPU).
- _Extensions to experiments and ablations (mvhb)_: We conduct ablation studies for ISRUC and FOG dataset, in addition to WESAD. We further conduct experiments using an additional skin conductance (SC) modality for the FOG dataset. We have also expanded our future work discussion to highlight how our proposed framework can be extended.

In addition, we provide additional analyses in response to the questions raised by reviewers:
- _Bridge position selection results_: We provide and analyze the detailed layer-wise linear probing and CKA similarity scores for selecting bridge input and output positions.
- _Bridge position impact on transfer performance_: We analyze how different bridge input and output positions impact unsupervised cross-modal knowledge transfer performance.
- _Additional ablations_: We study how the transfer performance changes when we replace the proposed BioX-Bridge architecture with a fully connected model. We further study how transfer performance varies when considering different subjects by reshuffling the subjects across splits.

We believe that the clarifications and additional analyses have strengthened the paper and resolved concerns from reviewers. In particular, reviewer iQEx responded positively to our rebuttal and kindly raised the score from 4 to 6. The modified scores for this paper were 8, 6, 6 on Nov 26, which was one day before the OpenReview software bug was made aware of.

In conclusion, we would like to emphasize the key contribution: a novel unsupervised model bridging framework that enables cross-modal knowledge transfer through information flow between biosignal models. This opens up the possibility of performing the same tasks using alternative biosignal modalities.

We hope this work lays the foundation towards more accessible, adaptable, and modality-agnostic health monitoring, where compute resources and labeled data are often limited.

---

### Meta-Review · Program_Chairs · 2026-01-08

**Summary:**

This paper a framework for unsupervised knowledge transfer across modalities between biosignal foundation models. The method proposes a parameter-efficient bridge network whose positions can be selected using linear probing and CKA similarity, and the method is backed up empirically on several examples. The paper shows promising performance and is written with clarity, but reviewers also find it lacking in justifying the methodology or containing sufficiently deep theoretical contributions, with more detail below. While the author response was thorough and appreciated ultimately there are enough of these core issues that would be more fully addressed in another round of review.

**Reviewer Concerns:**

Reviewers mention the lack of theoretical justificiation and I agree that this is a remaining concern. This includes ad hoc decisions for the per-token mapping constraint, or the alignment occurring at final layer only. Even though the bridge is a fairly lightweight implementation, it is unlikely users would know how to choose prototypes and how robust bridge positions are across datasets/how easy it is to calibrate associated parameters. The paper provides some insight and evidence that it works quite well on some tasks, but there is a gap in terms of a methodological core that would make the work more mature. Currently, probing all layers seems feasible for the pairwise modalities considered, but reviewers allude to the potential problem in growing complexity when one considers richer combinations of modalities.

**Reviewer Scores:**

Scores are mixed even after one negative reviewer agrees to slightly raise their score after response. The most positive review is unfortunately lacking meaningful detail from my view, though they acknowledged that they would stand by their score while also being open to either increasing or decreasing it.

---

### Decision · Program_Chairs · 2026-01-26

Accept (Oral)